# Microglia govern the extinction of acute stress-induced anxiety-like behaviors in male mice

Danyang Chen[1,6], Qianqian Lou[1,6], Xiang-Jie Song[1,6], Fang Kang[1], An Liu[2], Changjian Zheng[3], Yanhua Li[1], Di Wang[1], Sen Qun[4], Zhi Zhang [1,5] ✉, Peng Cao [4] ✉ & Yan Jin [4] ✉

Anxiety-associated symptoms following acute stress usually become extinct gradually within a period of time. However, the mechanisms underlying how individuals cope with stress to achieve the extinction of anxiety are not clear. Here we show that acute restraint stress causes an increase in the activity of GABAergic neurons in the CeA (GABA[CeA]) in male mice, resulting in anxiety-like behaviors within 12 hours; meanwhile, elevated GABA[CeA] neuronal CX3CL1 secretion via MST4 (mammalian sterile-20-like kinase 4)-NF-κB-CX3CL1 signaling consequently activates microglia in the CeA. Activated microglia in turn inhibit GABA[CeA] neuronal activity via the engulfment of their dendritic spines, ultimately leading to the extinction of anxiety-like behaviors induced by restraint stress. These findings reveal a dynamic molecular and cellular mechanism in which microglia drive a negative feedback to inhibit GABA[CeA] neuronal activity, thus facilitating maintenance of brain homeostasis in response to acute stress.

Anxiety caused by acute stress is likely an evolutionary adaptation for maintaining heightened vigilance and sustained attention to stimuli that require close attention or for avoiding repeated exposure to dangerous conditions[1–3]. It is well-known that the acute anxiety can gradually become extinct with the removal of stress stimuli. However, in this dynamic extinction process, the mechanisms of how individuals cope with acute stress and drive anxiety extinction to avoid the irreversible effects induced by persistent anxiety are still unclear.

Microglia and neurons in the brain can communicate bidirectionally. Upon receiving external stimuli, neurons respond first and mediate a cascade of responses throughout the brain, including modulating the activity of neural immune cells, such as microglia, via secretion of soluble factors (e.g., chemokines, cytokines, and

neurotransmitters)[4,5]. As resident macrophages in the brain, microglia perform essential functions required for the maintenance of brain homeostasis, for instance, removing dying neurons, pruning non-functional synapses, or producing ligands that support neuronal survival[6–8]. Microglial activation is also known to participate in triggering anxiety in animal models of chronic stress[9–12]. Despite several decades of intensive research focusing on neuronal remodeling in the brain after acute or chronic stress, our understanding of the dynamic mechanisms regulating neuron–microglia interactions, and their role in recovery from acute stress-induced anxiety, remains incomplete.

Acute or chronic exposure to stress can lead to several long-lasting adaptive changes in stress-sensitive brain regions such as the amygdala, which has been shown to undergo radical changes in

[1]Department of Anesthesiology, the First Affiliated Hospital of USTC, Hefei National Laboratory for Physical Sciences at the Microscale, Division of Life Sciences and Medicine, University of Science and Technology of China, Hefei 230026, China. [2]Department of Physiology, School of Basic Medical Sciences, Anhui Medical University, Hefei 230022, China. [3]Department of Anesthesiology, the First Affiliated Hospital of Wannan Medical College, Wuhu 241002, China. [4]Stroke Center and Department of Neurology, The First Affiliated Hospital of USTC, Division of Life Sciences and Medicine, University of Science and Technology of China, Hefei 230026, China. [5]The Center for Advanced Interdisciplinary Science and Biomedicine, Institute of Health and Medicine, Division of Life Sciences and Medicine, University of Science and Technology of China, Hefei 230026, China. [6]These authors contributed equally: Danyang Chen, Qianqian Lou, Xiang-Jie Song. ✉e-mail: zhizhang@ustc.edu.cn; pengcao@ustc.edu.cn; jinyan@ustc.edu.cn

function and structure[13–15]. Neuroimaging data have shown that neural activity is consistently elevated in the amygdala of patients with mood disorders, such as fear and depression[16,17]. In particular, the central nucleus of the amygdala (CeA) serves as the primary output nucleus for amygdala functions, and its increased activity is well-established as a potent driver of anxiety-like behaviors in mice[1,18–20]. Although CeA neuronal activity appears to participate in stress responses resulting in neuropsychiatric disorders, the molecular and cellular mechanisms of these CeA neuronal responses in the extinction of acute stress-induced anxiety remain unclear.

In this study, we demonstrate how acute stress induces microglial activation via MST4 (mammalian sterile-20-like kinase 4)-NF-κB-CX3CL1 signaling in a manner that later promotes the extinction of anxiety-like behaviors in mice. We find that reduced MST4 expression in GABAergic neurons in the CeA (GABA^CeA) contributes to the increased release of CX3CL1, inducing microglial activation, which in turn increases the engulfment of GABA^CeA neuronal spines, thereby reducing GABA^CeA neuronal activity and alleviating anxiety-like behaviors within 12 hours after acute restraint stress. These results support a positive regulatory role of microglia in the temporary and reversible activation of the response to acute stress and, thus, maintaining brain homeostasis in male mice. These findings provide a mechanistic basis for the extinction of anxiety-like behaviors and suggest an intervention strategy for treating the effects of stress-induced remodeling in the brain.

## Results

### GABA^CeA neuronal activity participates in the onset and extinction of anxiety-like behaviors in acute stress mice

To investigate the neural basis for an acute stress response, we induced anxiety-like behaviors in male mice through acute restraint stress for 2 hours (ARS-2h), as previously described[21–23]. Then, the open field test (OFT) and elevated plus maze (EPM) test were used to assess exploratory behaviors in mice at 0.5 h, 4 h, 8 h, and 12 h post-ARS-2h and in non-stressed controls (Fig. 1a). Significant changes in exploratory behaviors, which are closely associated with anxiety-like behaviors, indicated by less time spent in the central area of the OFT and fewer entries into the open arms of the EPM, were observed at 0.5 h/ 4 h/8 h post-stress induction in ARS-2h mice compared with control mice, while no significant difference was observed between control mice and ARS-2h mice at 12 h after restraint (Fig. 1b–e). In addition, there is no significant difference in distance traveled in OFT at 0.5 h/ 4 h/8 h/12 h post ARS-2h from that of corresponding control mice (Supplementary Fig. 1). These data showed that changes in exploratory behaviors caused by acute restraint stress are extinct by 12 h after stress induction.

The CeA plays a critical role in the generation of anxiety states[19,24]. We then examined GABA^CeA neuronal activity at 0.5 h/4 h/ 8 h/and 12 h post-ARS-2h in freely moving mice by in vivo multi-tetrode recordings (Fig. 1f). The characteristics of spike waveforms of GABA^CeA neurons were identified by using optogenetic tagging in GAD2-Cre mice (Supplementary Fig. 2). The results showed that the spontaneous spike firing rate was augmented in GABA^CeA neurons of ARS-2h mice compared with control mice but decreased over time until it returned to levels comparable with the non-stressed controls at 12 h after restraint, aligning well with the results of behavioral tests (Fig. 1g, h).

To further investigate how GABA^CeA neurons participated in exploratory behaviors, GABA^CeA neuronal activity was inhibited by local injection of AAV-DIO-hM4Di-mCherry into the bilateral CeA of GAD2-Cre mice (Fig. 1i and Supplementary Fig. 3a). At three weeks after virus expression, whole-cell patch-clamp recordings in acute brain slices revealed that the resting membrane potential ($V_{rest}$) was hyperpolarized in GABA^CeA neurons after perfusion with Clozapine-N-oxide (CNO) (10 μM) (Supplementary Fig. 3b, c),

indicating that the virus was successfully expressed in GABA^CeA neurons. In addition, in ARS-2h mice, chemogenetic inhibition of GABA^CeA neurons increased the time spent and entries in the central area of the OFT and open arms of the EPM (Fig. 1j). Moreover, non-stressed naïve mice with chemogenetic activation of GABA^CeA neurons displayed low level of exploratory behaviors (Supplementary Fig. 4). These results suggested that acute restraint stress increases GABA^CeA neuronal activity and induces anxiety-like behaviors in mice, while a decrease in this neuronal activity is accompanied by the extinction of anxiety-like behaviors.

### Microglial activation is involved in the extinction of anxiety-like behaviors following acute stress

Microglia serve as critical modulators of neuronal activity and associated behavioral responses in mice, including anxiety-like behaviors[25–27]. To examine whether changes in GABA^CeA neuronal activity over time are related to microglial activation in ARS-2h mice, we examined the activation of microglia in the CeA through immunofluorescence detection of the microglia-specific marker ionized calcium-binding adapter molecule 1 (Iba1). At 0.5 h and 8 h post-ARS-2h, the number of Iba1+ microglia were significantly higher in model mice than in corresponding control mice, but this change was reversed at 12 h post-ARS-2h (Fig. 2a, b). Since microglial morphology is correlated with their activation status[28], we analyzed microglial morphology with semi-automatic quantitative morphometric three-dimensional (3D) measurements, which revealed significantly shorter processes and decreased branch points at 0.5 h and 8 h post-ARS-2h compared with corresponding control mice (Fig. 2c, d). Notably, these activation phenotypes of microglia returned to normal levels at 12 h post-ARS-2h in model mice (Fig. 2b–d).

From a molecular standpoint, we also tested the levels of inflammatory marker MHCII and a panel of inflammatory factors, including TNF-α, IL-1β, and IL-6 in microglia. Immunofluorescence staining showed that the expression of MHCII increased in ARS-2h mice compared with controls at 0.5 h post-stress induction associated with microglial activation (Supplementary Fig. 5a, b). In addition, we isolated CeA microglia by fluorescence-activated cell sorting using Cx3cr1-GFP transgenic mice, which express a GFP label in microglial cells, and found that the mRNA of Tnf-α, Il-1β and Il-6 analyzed by qPCR were expressed at significantly higher levels in ARS-2h mice than those of non-stressed controls at 0.5 h post-stress treatment (Supplementary Fig. 5c–e). To further validate our observations of changes in microglial density, we stained Ki67 (a marker for cell proliferation) and found that an obvious Ki67 expression in CeA microglia of ARS-2h mice at 0.5 h and 8 h post treatment compared to that in controls, followed by a gradual decrease over time (Supplementary Fig. 6a). Moreover, TUNEL assays showed that apoptosis levels peaked at 8 h post-ARS-2h, then returned to baseline levels at 12 h post-ARS-2h treatment (Supplementary Fig. 6b). These results led us to hypothesize that the activation of microglia in the CeA produced by acute stress may be involved in the extinction of anxiety-like behaviors.

To examine whether microglial activation in the CeA is responsible for ARS-induced anxiety-like behaviors, intracranial microinfusion with minocycline, which has been widely used to inhibit microglial activity, were delivered in the bilateral CeA by cannula implantation (Fig. 2e). The model mice with pre-microinjection of minocycline still exhibited decreased the time spent and entries in the central area of the OFT and open arms of the EPM at 12 h post-ARS-2h, while ARS-2h mice pre-administered with an ACSF control treatment displayed high levels of exploratory behaviors at 12 h post-stress induction (Fig. 2f, g). It also deserves mention that low levels of exploratory behaviors were observable at 0.5 h post-ARS-2h in mice pre-treated with either minocycline or ACSF (Fig. 2f, g). Immunofluorescence staining for Iba1 verified that microglial activation in the CeA was significantly inhibited in mice pre-treated with minocycline before ARS-2h stress induction

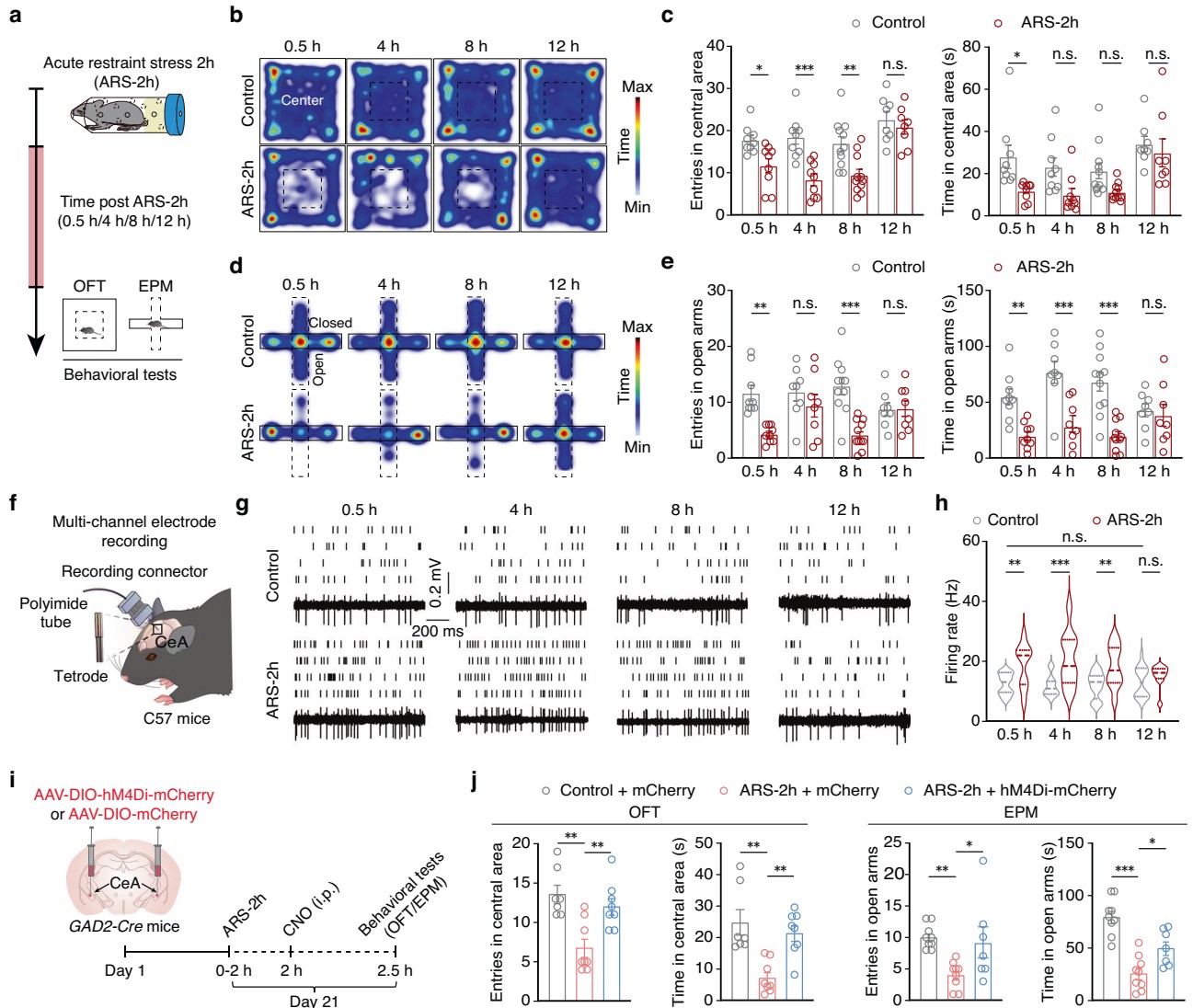

**Fig. 1 | GABA^CeA neurons mediate acute restraint stress-induced anxiety-like behaviors. a** Schematic of experimental design. Representative heatmaps (**b**) and summarized data of entries and the time spent in central area (**c**) of the open field test (OFT) from the indicated groups (0.5 h, $n = 9$ mice per group; 4 h, $n = 9$ mice per group; 8 h, $n = 11$ mice per group; 12 h, $n = 8$ mice per group). Representative heatmaps (**d**) and summarized data of entries and the time spent in the open arms (**e**) of the elevated plus maze (EPM) from the indicated groups (0.5 h, $n = 9$ mice per group; 4 h, $n = 8$ mice per group; 8 h, $n = 11$ mice per group; 12 h, $n = 8$ mice per group; **e** left, $F_{1,64} = 23.89$, $p < 0.001$; right, $F_{1,64} = 42.23$, $p < 0.001$). **f** Schematic of multi-channel electrode recordings. Raster plots and typical traces (**g**) and the quantitative data (**h**) of the spontaneous firings of GABA^CeA neurons from the indicated groups ($n = 20$ cells from six mice per group; **h** $F_{1,152} = 43.48$, $p < 0.001$).

**i** Schematic of the experimental procedure. **j** Summarized data of entries and the time spent in the central area and open arms of OFT and EPM from the indicated groups (OFT: Control + mCherry, $n = 7$ mice, ARS-2h + mCherry, $n = 8$ mice, ARS-2h + hM4Di, $n = 8$ mice; left, $F_{2,20} = 10.18$, $p = 0.0009$; right, $F_{2,20} = 10.68$, $p = 0.0007$; EPM: Control + mCherry, $n = 9$ mice, ARS-2h + mCherry, $n = 9$ mice, ARS-2h + hM4Di, $n = 7$ mice; left, $F_{2,22} = 6.391$, $p = 0.0065$; right, $F_{2,22} = 22.87$, $p < 0.001$). Significance was assessed by two-way repeated-measures ANOVA with post hoc comparison between groups in (**c, e, h**), one-way ANOVA with post hoc comparison between groups in (**j**). All data are presented as mean ± SEM. *$p < 0.05$, **$p < 0.01$, ***$p < 0.001$; n.s., not significant. See also Supplementary Data 1. Source data are provided as a Source Data file.

compared with mice pre-treated with ACSF (Supplementary Fig. 7). No apparent reactive gliosis was observed in the CeA of mice with cannula implantation (Supplementary Fig. 8). In addition, in vivo multitetrode recordings in freely moving ARS-2h mice with CeA pre-microinjection of minocycline showed that spontaneous neuronal activity of GABA^CeA neurons was significantly higher at 12 h post-ARS-2h compared with that in ARS-2h mice pre-administered saline (Fig. 2h, i), although no difference in spontaneous neuronal activity of GABA^CeA neurons was detected at 0.5 h between ARS-2h mice treated with minocycline and those treated with saline (Fig. 2h, i). These results indicated that inhibiting microglial activation blocks the recovery of GABA^CeA neuronal hyperactivity and the extinction of anxiety-like behaviors induced by acute stress.

## Microglial engulfment of GABA^CeA neuronal spines contributes to the extinction of anxiety-like behaviors induced by acute stress

Previous reports have shown that activated microglia can influence neuronal activity through direct engulfment of spines in the brain[29–32]. To investigate the potential relationship between microglial sensitivity to stress and GABA^CeA neuronal hypoactivity during the extinction of anxiety-like behaviors, we examined the remodeling of GABA^CeA neuronal spines following ARS-2h in model mice. To this end, we performed cell type-specific sparse labeling by injecting AAV-CSSP-YFP-8E3 virus into the bilateral CeA of *GAD2-Cre* mice to specifically label the dendritic spines of GABA^CeA neurons (Fig. 3a). Compared with corresponding control mice, the density of GABA^CeA neuronal dendritic

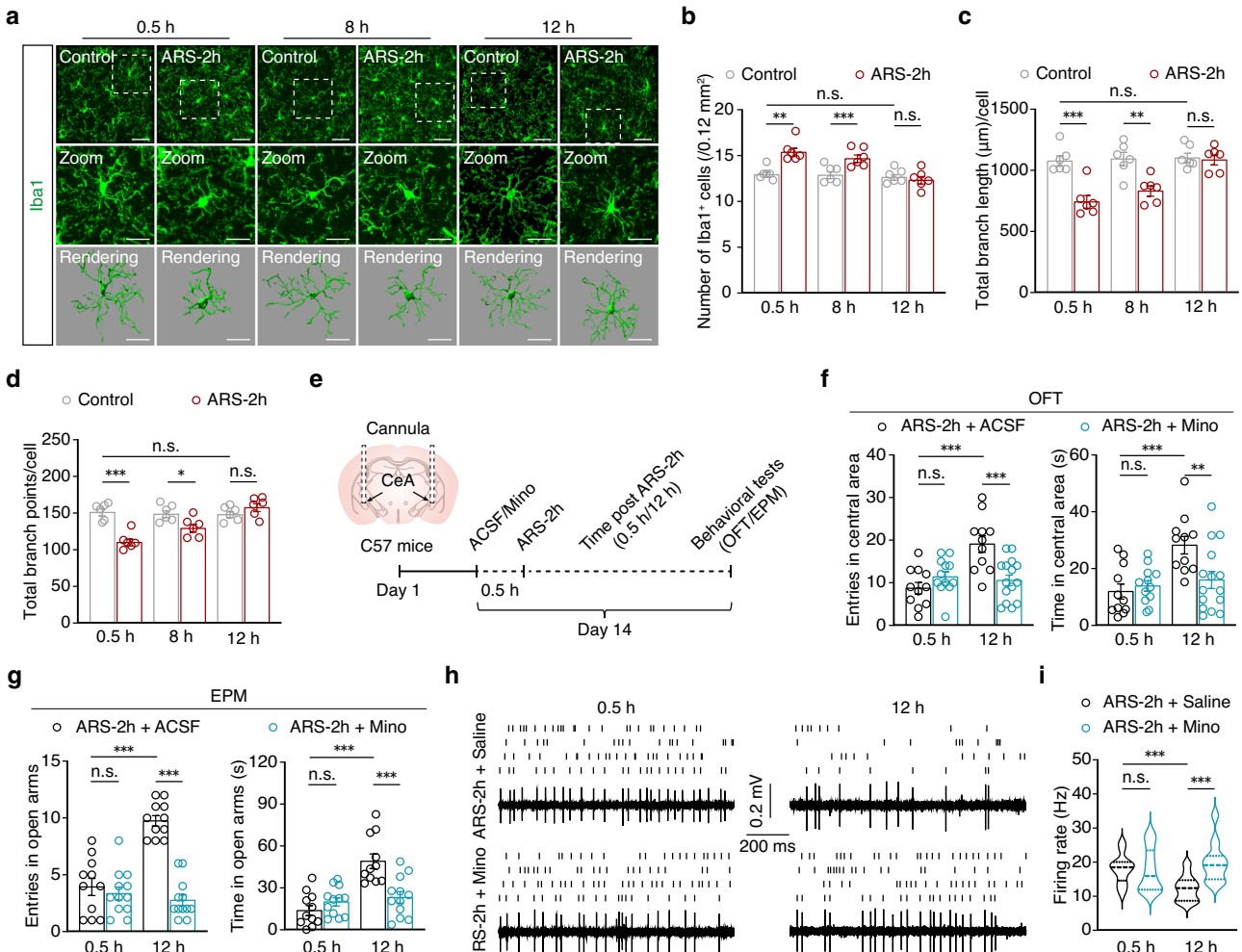

**Fig. 2 | Extinction of anxiety-like behaviors depends on activated microglia.**
**a** Representative images of Iba1 immunostaining and three-dimensional reconstruction of microglia in the CeA from the indicated groups. Scale bars, 40 μm (overview) and 20 μm (inset and rendering). **b** Quantification of Iba1$^+$ cell numbers in the CeA from the indicated groups ($n = 6$ mice per group; $F_{1,30} = 18.79$, $p = 0.0002$). IMARIS-based semi-automatic quantification of cell morphometry, including the total process length (**c**) and number of branch points (**d**) of Iba1$^+$ microglia in the CeA from the indicated groups ($n = 6$ mice per group; **c** $F_{1,30} = 29.49$, $p < 0.001$; **d** $F_{1,30} = 18.67$, $p = 0.0002$). **e** Experimental scheme of ARS-2h mice pre-treated with ACSF or minocycline (Mino). **f** Summarized data of entries and the time spent in central area of OFT (0.5 h: ARS-2h + ACSF, $n = 11$ mice, ARS-2h

+ Mino, $n = 12$ mice; 12 h: ARS-2h + ACSF, $n = 11$ mice, ARS-2h + Mino, $n = 15$ mice; left, $F_{1,45} = 4.068$, $p = 0.0497$; right, $F_{1,45} = 3.477$, $p = 0.0688$). **g** Summarized data of entries and the time spent in central area of EPM (0.5 h: ARS-2h + ACSF, $n = 11$ mice, ARS-2h + Mino, $n = 12$ mice; 12 h: ARS-2h + ACSF, $n = 11$ mice, ARS-2h + Mino, $n = 12$ mice; left, $F_{1,42} = 43.09$. $p < 0.001$; right, $F_{1,42} = 6.041$, $p = 0.0182$). Raster plots and typical traces (**h**) and the quantitative data (**i**) of the spontaneous firings of GABA$^{CeA}$ neurons from the indicated groups ($n = 24$ cells from six mice per group; **i** $F_{1,92} = 12.86$, $p = 0.0005$). Significance was assessed by two-way repeated-measures ANOVA with post hoc comparison between groups in (**b, c, d, f, g, i**). All data are presented as mean ± SEM. *$p < 0.05$, **$p < 0.01$, ***$p < 0.001$; n.s., not significant. See also Supplementary Data 1. Source data are provided as a Source Data file.

spines was increased at 0.5 h and 8 h post-ARS-2h in model mice but recovered to control levels by 12 h post-ARS-2h (Fig. 3b, c). However, GABA$^{CeA}$ neuronal dendritic spine density was not significantly different between model mice pre-treated with minocycline (i.p.) and those treated with saline at 0.5 h post-ARS-2h, but was markedly higher in the minocycline pre-treatment group at 12 h post-stress induction compared with the saline pre-treatment group in model mice (Fig. 3d–f). These results suggested that enhanced microglial engulfment is required for the observed reduction in dendritic spine density of GABA$^{CeA}$ neurons during the extinction of anxiety-like behaviors induced by acute restraint stress.

To investigate the interactions between microglial processes and dendritic spines of GABA$^{CeA}$ neurons, we examined the phagocytic function of microglia in ARS-2h mice. Immunofluorescence staining for the macrophage marker CD68 showed significantly higher levels in ARS-2h mice compared with controls at 0.5 h post-stress induction, but no detectable difference between stressed and control mice at 12 h

post-ARS-2h, consistent with observed changes in microglial Iba1 expression (Fig. 3g). In addition, quantitative engulfment assays showed that microglia contained more YFP$^+$ neuronal dendritic spines, accompanied by greater microglia–dendrite contact areas, in ARS-2h mice than those in control animals at 0.5 h, but not at 12 h, post-stress induction (Supplementary Fig. 9). Moreover, 3D reconstruction showed a significant increase in the colocalization of GAD65/67 immunoreactive puncta, a GABAergic neuron-specific marker, and Iba1$^+$ microglia in the CeA at 0.5 h post-ARS-2h, compared with control mice, and this phenomenon no longer occurred at 12 h post-ARS-2h in model mice (Fig. 3h). Confocal imaging data and 3D surface rendering further depicted abundant colocalization between immunoreactive puncta of GAD65/67, CD68, and Iba1$^+$ microglia in the CeA of ARS-2h mice at 0.5 h, but not at 12 h post-treatment, nor in the corresponding control animals (Fig. 3i). Following minocycline pre-treatment in ARS-2h mice, microglial phagocytosis of GAD65/67 signals in the CeA was significantly lower than that in the saline pre-treatment group (Fig. 3j).

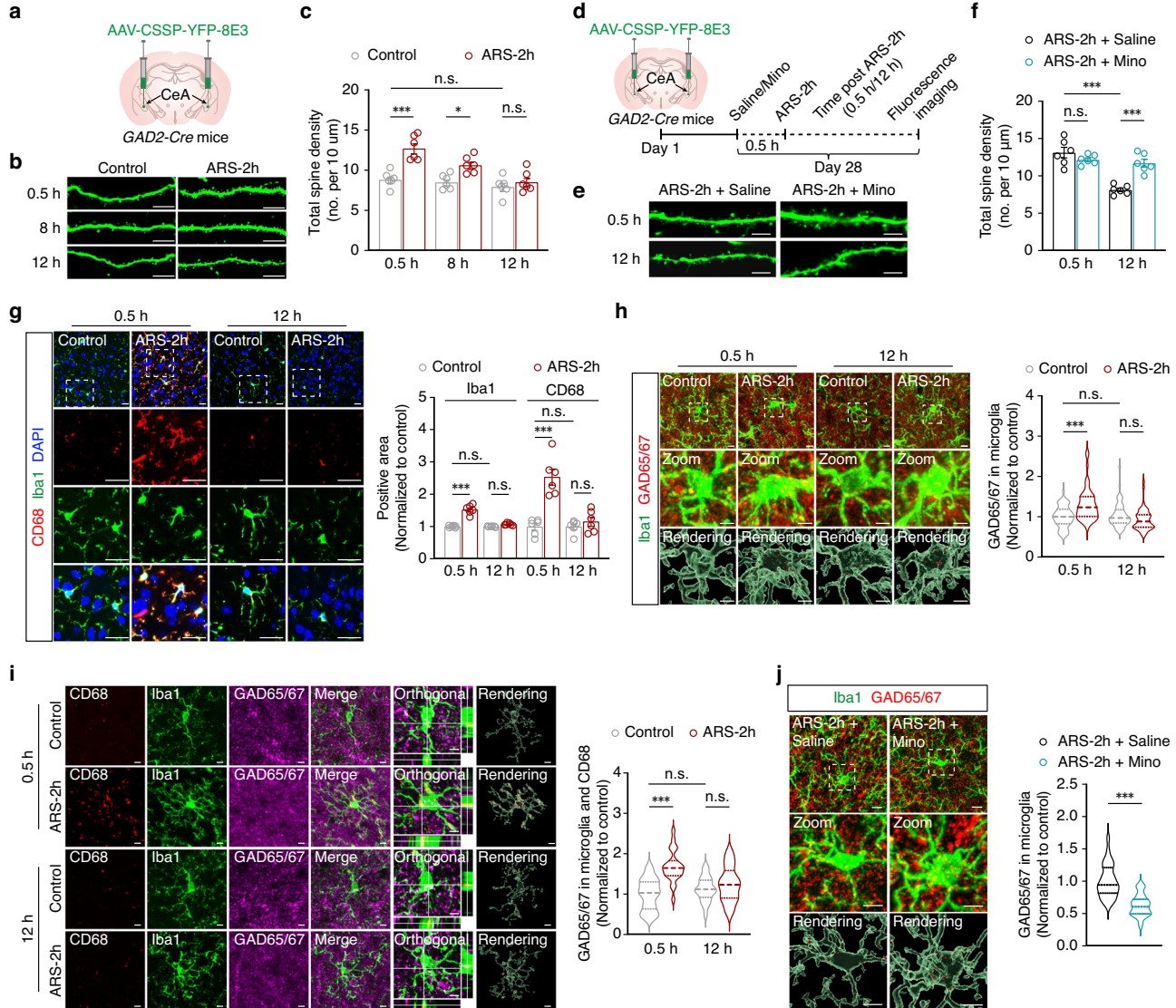

**Fig. 3 | Extinction of anxiety-like behaviors via microglial engulfment of GABA<sup>CeA</sup> dendritic spines.** **a** Schematic of bilateral virus infection. Representative images of neuronal dendrites (**b**) and quantification of spine numbers per 10 μm (**c**) ($n = 6$ mice per group; $F_{1,30} = 31.40$, $p < 0.001$). Scale bars, 10 μm. **d** Experimental scheme for virus injection. Representative images of neuronal dendrites (**e**) and quantification of spine numbers (**f**) from the indicated groups ($n = 6$ mice per group; **f** $F_{1,20} = 7.743$, $p = 0.0115$). Scale bars, 10 μm. **g** Representative images (left) and quantitative analyses (right) of immunostaining for CD68 (red), Iba1 (green), and DAPI (blue) in the CeA from the indicated groups ($n = 6$ mice per group; Iba1, $F_{1,20} = 91.50$, $p < 0.001$; CD68, $F_{1,20} = 30.28$, $p < 0.001$). Scale bars, 20 μm. **h** Representative images and three-dimensional surface rendering (left) of Iba1+ microglia (green) containing GAD65/67+ puncta (red) and DAPI (blue) in the CeA from the indicated groups. Quantification (right) of GAD65/67<sup>+</sup> puncta in microglia

in mice ($n = 53$ cells from six mice per group; $F_{1,208} = 4.789$, $p = 0.0298$). Scale bars, 10 μm (overview) and 5 μm (inset and rendering). **i** Representative images (left) and quantitative analyses (right) of immunostaining for CD68 (red), Iba1 (green), and GAD65/67<sup>+</sup> puncta (purple) in the CeA from the indicated groups ($n = 24$ cells from six mice per group; $F_{1,92} = 26.55$, $p < 0.001$). Scale bars, 5 μm. **j** Representative images (left) and quantification (right) of Iba1<sup>+</sup> microglia (green) containing GAD65/67<sup>+</sup> puncta (red) in the CeA from the indicated groups ($n = 30$ cells from six mice per group; $t_{58} = 6.739$, $p < 0.001$). Scale bars, 10 μm (overview) and 5 μm (inset and rendering). Significance was assessed by two-way repeated-measures ANOVA with post hoc comparison between groups in (**c, f, g, h, i**), two-tailed unpaired Student's $t$-test in (**j**). All data are presented as mean ± SEM. *$p < 0.05$, ***$p < 0.001$; n.s., not significant. See also Supplementary Data 1. Source data are provided as a Source Data file.

These results together suggested that activated microglia can inhibit GABA<sup>CeA</sup> neuronal activity through an apparent negative feedback mechanism following acute restraint stress to promote the recovery from anxiety-like behaviors in the later stage.

## Hyperactivity of GABA<sup>CeA</sup> neurons activates microglia in the CeA of acute stress mice

To better understand how GABA<sup>CeA</sup> neurons and microglial cells respond to acute stress stimuli, we examined GABA<sup>CeA</sup> neuronal activity and microglial morphology in mice with acute stress induced by ARS-5min and ARS-30min. The results showed that the spontaneous

firing rate of GABA<sup>CeA</sup> neurons was significantly increased after 5 min of restraint and that this increase also appeared after 30 min of restraint stress compared with control mice (Supplementary Fig. 10a, b). Furthermore, although the number of Iba1<sup>+</sup> microglia did not change in either ARS-5min or ARS-30min mice compared that with controls, the branch length and points of microglia were significantly lower at ARS-30min, and these changes were not observable in ARS-5min mice compared with controls (Supplementary Fig. 10c, d). These collective findings suggested that activation of GABA<sup>CeA</sup> neurons precedes microglial activation in the CeA of mice treated with acute restraint stress.

Next, we further investigated whether GABA^CeA neuronal hyperactivity is causally related to microglial activation in acute restraint stress mice. We found that chemogenetic activation of GABA^CeA neurons in naïve mice led to microglial activation, indicated by an increased number of Iba1+ microglia and decreased branch length and points in these cells (Supplementary Fig. 11a, b). In addition, chemogenetic inhibition of GABA^CeA neurons abolished the observed activation of microglia in ARS-2h mice (Supplementary Fig. 11c, d). Taken together, these results suggested that microglial activation results from elevated GABA^CeA neuronal activity induced by acute restraint stress in mice.

## CX3CL1-CX3CR1 signaling contributes to the extinction of anxiety-like behaviors following acute stress

To further explore potential molecular mechanisms through which microglial activation induced by hyperactivity of GABA^CeA neurons following acute stress, we used qPCR to screen for expression of a number of molecules previously reported to be involved in the response to stress and to mediate neuronal-microglia interactions[33–37]. We found that the mRNA levels of *Cx3cl1*, *Ccl2*, *Il-1β*, *Tnf-α*, and *Il-6* were significantly higher in samples of the CeA tissues collected at 0.5 h post treatment from ARS-2h than those in control mice (Supplementary Fig. 12). It is well known that CX3CL1/fractalkine, which is a secreted chemokine specifically expressed in neurons and widely involved in activity-dependent synaptic pruning, can be cleaved to yield a soluble isoform that triggers microglial activation through binding to its receptor, CX3CR1[38]. We, therefore, focused on the CX3CL1-CX3CR1 signaling in the regulation of the interactions between neurons and microglia. As expected, Western blot assays showed that the level of soluble CX3CL1 protein was indeed significantly higher at 0.5 h and 8 h in ARS-2h mice than that in control mice (Fig. 4a, b).

To investigate whether this enhanced soluble CX3CL1 in the CeA was involved in regulating microglial activation, we injected the CX3CR1-specific antagonist JMS-17-2 or the vehicle control in ARS-2h mice. Microglial activation, indicated by Iba1 expression, in the CeA was significantly inhibited in JMS-17-2 pre-treated acute restraint stress mice compared with those given the vehicle (Fig. 4c, d). In addition, immunofluorescence staining further illustrated that microglial phagocytosis of GAD65/67 puncta in the CeA decreased significantly in ARS-2h mice pretreated with the CX3CR1 antagonist compared with those pre-treated with vehicle (Fig. 4e, f). Next, we examined whether the change in GABA^CeA neuronal dendritic spine density was also affected by the CX3CR1 antagonist in ARS-2h mice. GABA^CeA neuron-specific expression of AAV-CSSP-YFP-8E3 virus in *GAD2-Cre* mice showed no difference in the dendritic spine density of these neurons at 0.5 h post-ARS-2h between JMS-17-2- and vehicle-pretreated model mice (Fig. 4g, h), while the spine density was obviously greater in these neurons at 12 h post-ARS-2h following pre-treatment with JMS-17-2 compared with the vehicle group after acute restraint stress (Fig. 4g, h).

To examine whether this enhanced soluble CX3CL1 in the CeA is responsible for ARS-induced anxiety-like behaviors, we bilaterally injected the JMS-17-2, or the vehicle control, into the CeA of ARS-2h mice. We found that mice pre-treated with JMS-17-2 showed significantly lower exploratory behaviors than vehicle-treated mice at 12 h post-ARS-2h, indicated by fewer entries and less time spent in the central area of the OFT and open arms of the EPM (Fig. 4i, j), which were still observed in both the JMS-17-2 and vehicle groups at 0.5 h post-ARS-2h (Fig. 4i, j). Moreover, in vivo multitetrode recordings in freely moving ARS-2h mice showed no difference in GABA^CeA neuronal spontaneous firing activity between the JMS-17-2 and vehicle-treated mice at 0.5 h post-ARS-2h, which was consistent with the results of behavioral tests; however, GABA^CeA neuronal firing activity was higher at 12 h post-stress induction in mice with the CeA pre-injection of JMS-17-2 than in vehicle controls (Fig. 4k, l).

To explore how *Cx3cl1* mRNA increased by ARS-2h, we used qPCR to examine *Nf-κb* mRNA expression, which is a transcription factor for mediating the expression of cytokines, including CX3CL1[39]. Interestingly, we found that *Nf-κb* mRNA levels were significantly increased in the CeA at 0.5 h post-ARS-2h compared with non-stressed control mice (Fig. 4m). Next, we administrated Pyrrolidinedithiocarbamate ammonium (PDTC), a selective inhibitor of NF-κB, into the CeA. In these mice, the increased time and entries in the central area of the OFT and open arms of the EPM at 12 h post-ARS-2h was prevented following PDTC administration (Fig. 4n, o). In addition, a significant decrease in *Cx3cl1* mRNA expression was observed in PDTC-treated model mice at 0.5 h post-ARS-2h compared to that in vehicle control animals (Fig. 4p). These findings indicated that the NF-κB signaling pathway is likely involved in regulating CX3CL1 protein expression in acute stress states.

Together, these findings suggest that, within 12 h after stress induction, elevated soluble CX3CL1 protein can promote microglial phagocytosis of GABA^CeA neuronal spines, thereby reducing GABA^CeA neuronal activity and driving the recovery from anxiety-like behaviors in model mice.

## GABA^CeA neuronal MST4-NF-κB-CX3CL1 signaling initiates acute stress-induced microglial activation

In light of the above mentioned evidence supporting the role of ARS-induced NF-κB-CX3CL1 signaling upregulation in the CeA, we next examined the mechanisms potentially responsible for initiating this process in ARS. The kinase MST4 has been previously reported to directly phosphorylate TRAF6 (TNF receptor-associated factor 6), inhibiting its ubiquitination, and subsequently limiting inflammatory response[40]. It thus has been considered as a "brake" on TLR-TRAF6-NF-κB-mediated inflammatory responses. To test whether MST4 participates in microglial activation in ARS, we conducted immunofluorescence staining of MST4 in the brain and found that it was widely distributed in several regions, including the accumbens nucleus, caudate putamen, paraventricular hypothalamic nucleus, and CeA (Supplementary Fig. 13, and Fig. 5a). We also found that MST4 predominantly colocalized with GABA-specific antibody, but not microglia, in the CeA (Supplementary Fig. 14). qRT-PCR and Western blot assays showed that MST4 expression was significantly decreased at both the mRNA and protein levels in the CeA at 0.5 h post-ARS-2h compared with non-stressed control mice (Fig. 5b, c), and returned to that of controls at 12 h post-ARS-2h (Fig. 5c).

To further explore MST4 function in the regulation of GABA^CeA neuronal activity and exploratory behaviors in mice, we next generated a Cre-dependent AAV vector to overexpress MST4 in GABA^CeA neurons of *GAD2-Cre* mice. Three weeks after bilateral CeA injection with rAAV-Ef1α-DIO-MST4-P2A-mCherry-WPRE-pA (AAV-MST4) or rAAV-Ef1α-DIO-mCherry-WPRE-pA (AAV-mCherry) viruses in *GAD2-Cre* mice (Fig. 5d), fluorescence microscopy showed that the mCherry+ signal was largely co-localized with the signal from GABA-specific antibody in the CeA of naïve mice (Fig. 5e). Western blots showed significantly greater MST4 expression in GABA^CeA neurons of naïve mice infected with AAV-MST4 than in control mice infected with AAV-mCherry (Fig. 5f). We then examined exploratory behaviors in *GAD2-Cre* mice injected with either AAV-MST4 or AAV-mCherry viruses (Fig. 5g) and found that GABA^CeA neuron-specific MST4 overexpression, but not the AAV-mCherry control, could lead to increasing time and entries in the central area of the OFT and open arms of the EPM in ARS-2h mice (Fig. 5h, i). In addition, GABA^CeA neuronal activity was restored in AAV-MST4-infected ARS mice (Fig. 5j, k), accompanied by lower mRNA levels of *Nf-κb* and *Cx3cl1* in the CeA (Fig. 5l), compared with AAV-mCherry-infected ARS mice. Moreover, microglial activation was significantly reversed in the CeA of ARS mice injected with AAV-MST4, with markedly fewer Iba1+ cells and increased total microglial branch length and points compared with ARS-2h mice with CeA-specific expression of the AAV-mCherry vector (Fig. 5m, n).

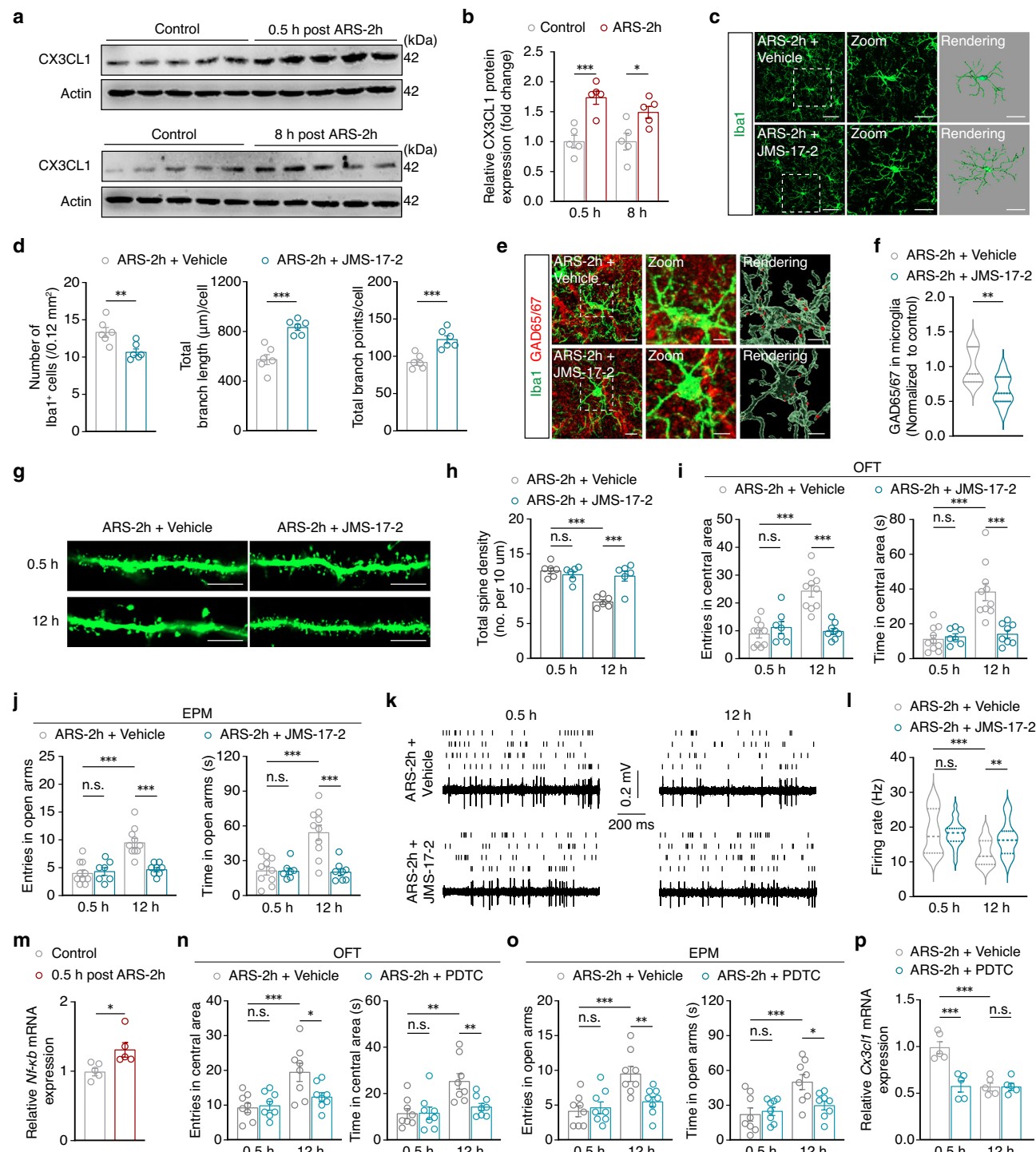

**Fig. 4 | Blocking CX3CL1-CX3CR1 signaling in the CeA prevents the extinction of anxiety-like behaviors following acute stress.** Representative images (**a**) and quantitative analyses (**b**) for Western blot of CX3CL1 in the CeA ($n = 5$ mice per group). **c** Representative images of Iba1 immunostaining and three-dimensional reconstruction of microglia in the CeA. Scale bars, 40 μm (overview) and 20 μm (inset and rendering). **d** The ARS-treated mice showed inhibited microglial activation after treated with JMS-17-2 ($n = 6$ mice per group). Representative images (**e**) and quantification (**f**) of Iba1+ microglia (green) containing GAD65/67+ puncta (red) in the CeA ($n = 10$ cells from five mice per group). Scale bars, 10 μm (overview) and 5 μm (inset and rendering). Representative images of neuronal dendrites (**g**) and quantification of spine numbers (**h**) ($n = 6$ mice per group; h, $F_{1,20} = 10.16$, $p = 0.0046$). Scale bars, 10 μm. Summarized data of entries and the time spent in

the central area and open arms of OFT (**i**) and EPM (**j**) (**i** left, $F_{1,31} = 11.51$, $p = 0.0019$; right, $F_{1,31} = 11.54$, $p = 0.0019$; **j** left, $F_{1,31} = 11.24$, $p = 0.0021$; right, $F_{1,31} = 13.51$, $p = 0.0009$). Raster plots and typical traces (**k**) and the quantitative data (**l**) of the spontaneous firings of CeA^GABA neurons ($n = 27$ cells from six mice per group; l, $F_{1,104} = 3.214$, $p = 0.0759$). **m** qPCR of *Nf-κb* in the CeA ($n = 5$ mice per group). Summarized data of entries and the time spent in the central area and open arms of OFT (**n**) and EPM (**o**) ($n = 8$ mice per group). **p** qPCR of *Cx3cl1* in the CeA ($n = 5$ mice per group; $F_{1,16} = 15.63$, $p = 0.0011$). Significance was assessed by two-way repeated-measures ANOVA with post hoc comparison between groups in (**b**, **h**, **i**, **j**, **l**, **n**, **o**, **p**), two-tailed unpaired Student's *t*-test in (**d**, **f**, **m**). All data are presented as mean ± SEM. *$p < 0.05$, **$p < 0.01$, ***$p < 0.001$; n.s., not significant. See also Supplementary Data 1. Source data are provided as a Source Data file.

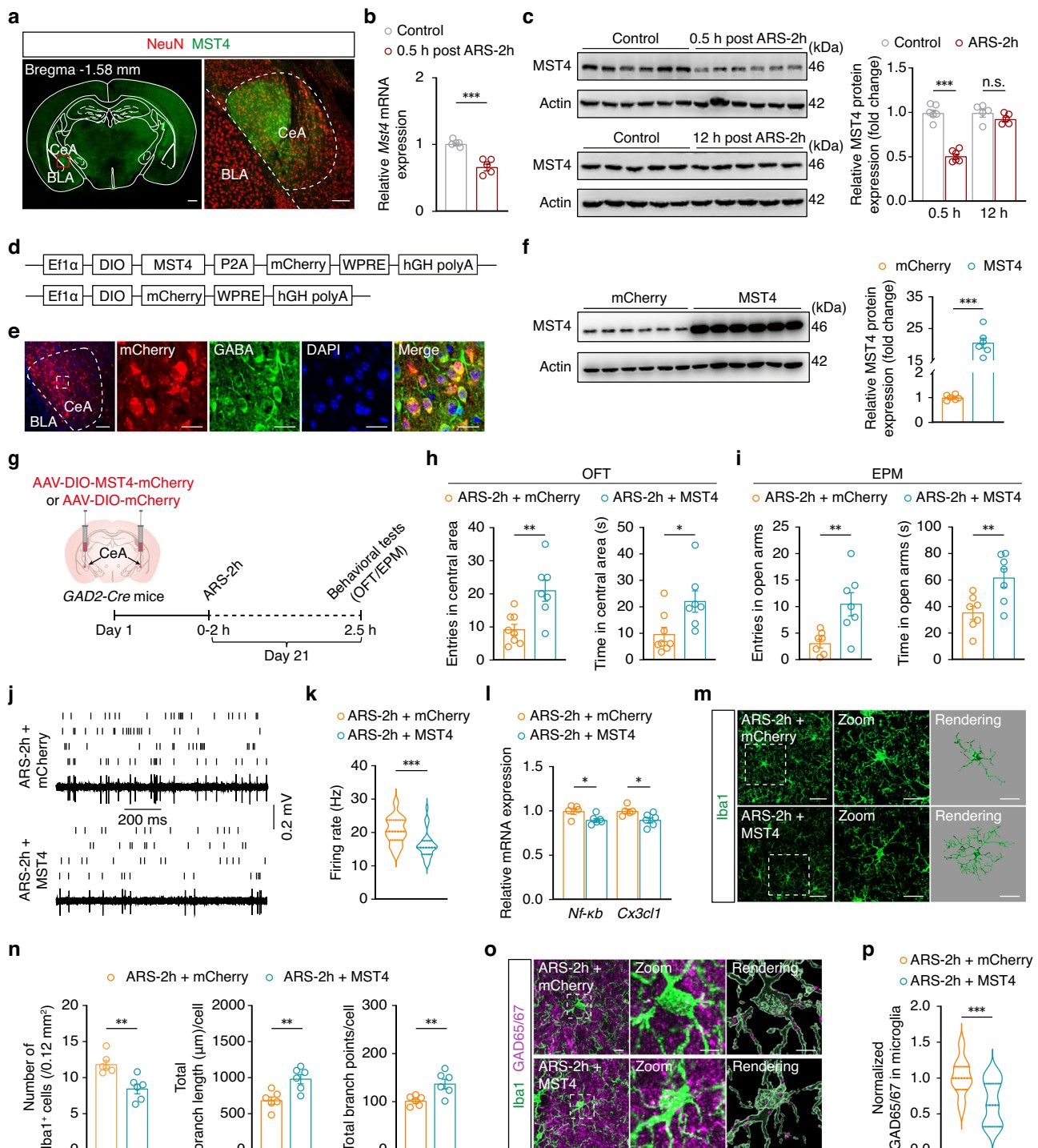

**Fig. 5 | Overexpression of MST4 in GABA^CeA neurons alleviates anxiety-like behaviors induced by acute stress. a** Immunofluorescence staining of MST4 (green) and NeuN (red) in the CeA. Scale bars, 500 μm (left) and 100 μm (right). **b** qPCR of *Mst4* in the CeA ($n = 5$ mice per group). **c** Western blot analysis of MST4 in the CeA (0.5 h, $n = 6$ mice per group; 12 h, $n = 5$ mice per group; $F_{1,18} = 69.87$, $p < 0.001$). **d** Schematic illustration of the MST4 overexpression construction strategy. **e** Representative images showing mCherry with co-labeling of GABA-specific antibody. The experiment was repeated three times. Scale bars, 100 μm (left) and 20 μm (right). **f** Western blot analysis of MST4 in the CeA ($n = 6$ mice per group; $t_{10} = 12.34$, $p < 0.001$). **g** Schematic of the experimental procedure. Summarized data of entries and the time spent in the central area and open arms of OFT (**h**) and EPM (**i**). Raster plots and typical traces (**j**) and the quantitative data (**k**) of the spontaneous firings of CeA^GABA neurons ($n = 30$ cells from six mice per group;

**k** $t_{58} = 4.203$, $p < 0.001$). **l** qPCR of *Nf-κb* and *Cx3cl1* in the CeA from the indicated groups ($n = 6$ mice per group). **m** Representative images of Iba1 immunostaining and three-dimensional reconstruction of microglia in the CeA. Scale bars, 40 μm (overview) and 20 μm (inset and rendering). **n** The ARS-treated mice showed inhibited activation of microglia after overexpression of MST4 in the CeA ($n = 6$ mice per group). Representative images (**o**) and quantification (**p**) of Iba1⁺ microglia (green) containing GAD65/67⁺ puncta (purple) in the CeA ($n = 18$ cells from six mice per group). Scale bars, 10 μm (overview) and 5 μm (inset and rendering). Significance was assessed by two-way repeated-measures ANOVA with post hoc comparison between groups in (**c**), two-tailed unpaired Student's *t*-test in (**b, f, h, i, k, l, n, p**). All data are presented as mean ± SEM. *$p < 0.05$, **$p < 0.01$, ***$p < 0.001$. See also Supplementary Data 1. Source data are provided as a Source Data file.

Notably, microglial engulfment of GAD65/67 puncta was also significantly decreased in acute restraint stress mice with GABA$^{CeA}$ neurons infected with AAV-MST4 compared with those with AAV-mCherry infection (Fig. 5o, p). It should also be mentioned that naïve mice overexpressing MST4 in GABA$^{CeA}$ neurons exhibited no changes in exploratory behaviors and showed no significant difference in GABA$^{CeA}$ neuronal activity compared with mice expressing control virus (Supplementary Fig. 15).

To explore the effects of MST4 signaling on microglial activation in the CeA, we next generated a Cre-dependent AAV vector expressing MST4 short-hairpin RNAs (shRNAs) to knock down MST4 expression. At three weeks after CeA injection with rAAV-CMV-DIO-mCherry-U6-

shRNA (MST4) (AAV-shMST4) or rAAV-CMV-DIO-mCherry-U6-shRNA (scramble) (AAV-mCherry) in naïve *GAD2-Cre* mice (Fig. 6a), immunofluorescence staining showed that mCherry$^+$ cells were invariably co-labeled with GABA-specific antibody in the CeA (Fig. 6b). qPCR and Western blot assays showed that *Mst4* mRNA levels and MST4 protein levels significantly decreased, while *Nf-κb* and *Cx3cl1* mRNA levels and CX3CL1 protein expression increased in GABA$^{CeA}$ neurons, in naïve MST4 knockdown mice compared with mice infected with the AAV-mCherry control (Fig. 6c–e). Furthermore, microglia were activated and microglial engulfment of GAD65/67 was significantly increased in naïve mice with CeA injection of AAV-shMST4 compared with the AAV-mCherry control group (Fig. 6f–i). These results suggested that the

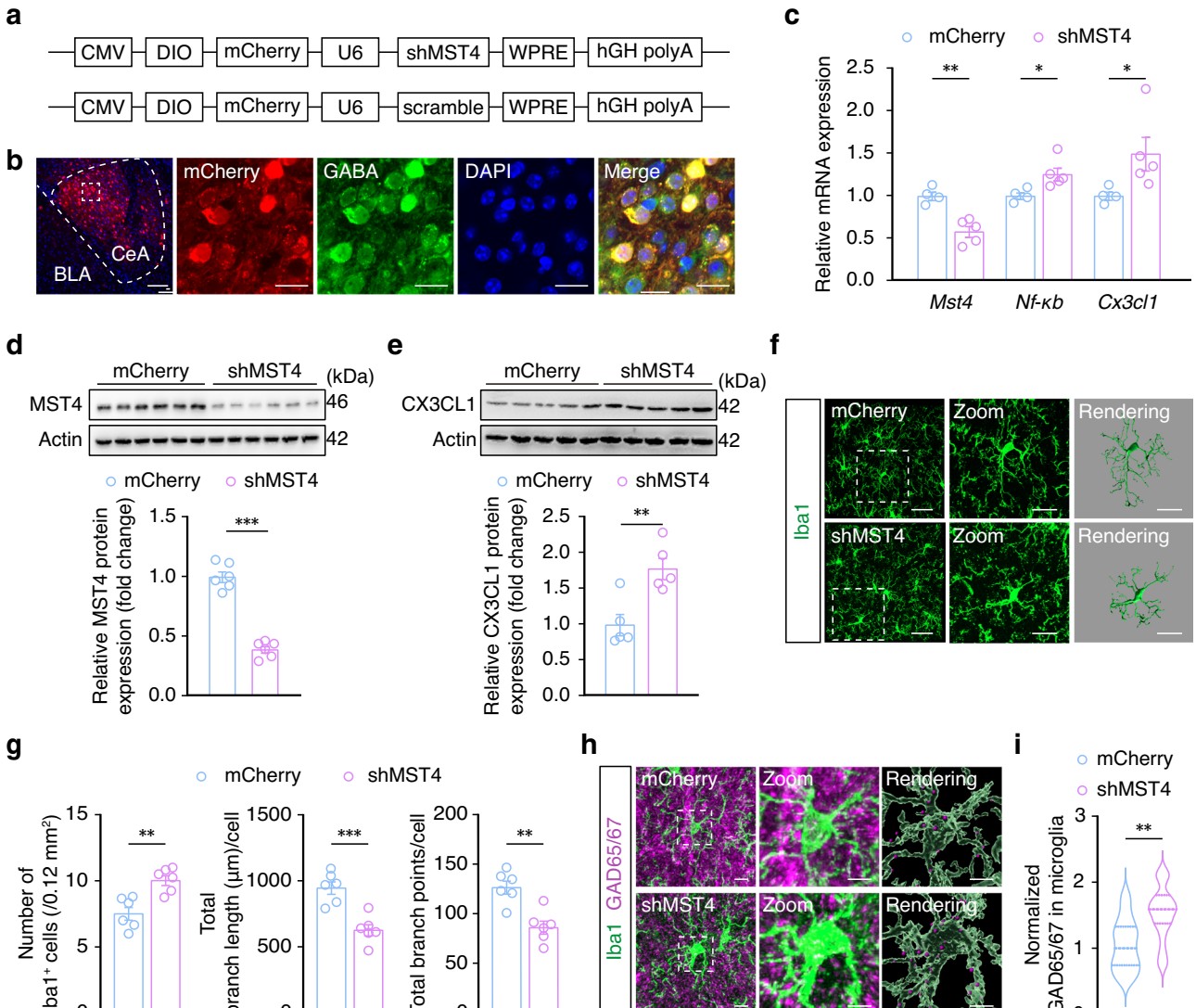

**Fig. 6 | MST4 knockdown in GABA$^{CeA}$ neurons activates microglia and increases the engulfment of GABA$^{CeA}$ neuronal spines. a** Schematic illustration of the construction strategy used for MST4 knockdown. **b** Representative images showing mCherry expression in neurons with co-labeling of GABA-specific antibody. These experiments have been repeated three times. Scale bars, 100 µm (left) and 20 µm (right). **c** qPCR of *Mst4*, *Nf-κb*, and *Cx3cl1* mRNA expression in the CeA of naïve mice infected with AAV-mCherry and AAV-shMST4 (mCherry, $n = 4$ mice per group, shMST4, $n = 5$ mice per group). **d** Western blot analysis of MST4 protein in the CeA of naïve mice infected with AAV-mCherry and AAV-shMST4 ($n = 6$ mice per group; $t_{10} = 11.73$, $p < 0.001$). **e** Western blot analysis of CX3CL1 protein in the CeA of naïve mice infected with AAV-mCherry and AAV-shMST4 ($n = 5$ mice per group). **f** Representative images of Iba1 immunostaining and three-dimensional

reconstruction of microglia in the CeA of naïve mice infected with AAV-mCherry and AAV-shMST4. Scale bars, 40 µm (overview) and 20 µm (inset and rendering). **g** Quantification of Iba1$^+$ cell numbers, total process length and number of branch points of microglia in the CeA from naïve mice infected with AAV-mCherry and AAV-shMST4 ($n = 6$ mice per group). Representative images (**h**) and quantification (**i**) of Iba1$^+$ microglia (green) containing GAD65/67$^+$ puncta (purple) in the CeA from naïve mice infected with AAV-mCherry and AAV-shMST4 ($n = 18$ cells from six mice per group). Scale bars, 10 µm (overview) and 5 µm (inset and rendering). Significance was assessed by two-tailed unpaired Student's *t*-test in (**c**, **d**, **e**, **g**, **i**). All data are presented as mean ± SEM. *$p < 0.05$, **$p < 0.01$, ***$p < 0.001$. See also Supplementary Data 1. Source data are provided as a Source Data file.

downregulation of MST4 could directly activate microglia and increase the engulfment of GABA<sup>CeA</sup> neuronal spines.

These cumulative findings suggest that a decrease in MST4 expression after acute restraint stress results in the activation of NF-κB-CX3CL1 signaling in GABA<sup>CeA</sup> neurons, which in turn promotes microglial activation and engulfment of GABA<sup>CeA</sup> neuronal spines, leading to the recovery of GABA<sup>CeA</sup> neuronal hyperactivity and the extinction of anxiety-like behaviors.

## Discussion

The findings of the current study suggest that the anxiety-like behaviors induced by acute restraint stress are extinct 12 h after stress induction in male mice and that the activity of GABA<sup>CeA</sup> neurons is increased but returns to non-stress levels. This process is mediated by microglial activation in the CeA and enhanced microglial engulfment of GABA<sup>CeA</sup> neuronal spines in acute restraint stress mice. Our results further reveal that MST4-NF-κB-CX3CL1 signaling in GABA<sup>CeA</sup> neurons mediates microglial activation (Supplementary Fig. 16). These cumulative findings suggest that a microglia-driven negative feedback mechanism, which operates similarly to inhibitory neurons, is important for protecting the brain from excessive activation of neurons in health or disease states.

Acute or chronic stress is associated with a higher risk of many psychological disorders, including anxiety and depressive disorders[41,42]. As the CeA is the main information output nucleus in the amygdala, many studies have reported that CeA hyperactivity following exposure to stress or stressor events contributes to the pathogenesis of anxiety[23,43,44]; these findings are consistent with our results. By contrast, here, we further explored a time-dependent correlation between GABA<sup>CeA</sup> neuronal activity and the onset and extinction of anxiety-like behaviors after acute restraint stress in mice. Notably, in addition to the hyperactivity of GABA<sup>CeA</sup> neurons, microglial activation in the CeA was also observed in mice during acute restraint stress treatment, which was dependent on the activation of GABA<sup>CeA</sup> neurons due to chemogenetic inhibition of these neurons reversing their activation.

Multiple studies have suggested that activation of microglia, immune monitoring cells in the brain, is involved in mediating responses to various stresses that can serve as a major trigger for many neuropsychiatric disorders[11,12,45]. In fact, microglia could be activated rapidly when the body suffers acute stress, indicated by changed morphology, enhanced phagocytic capacity, and an altered transcriptional profile, to detect changes in neuronal activity in real time and to promote the restoration of the brain homeostasis[7,46]. However, microglial immune function often becomes impaired by chronic stress, resulting in overproduction of inflammatory mediators and exacerbated phagocytic activity, which can be detrimental to the remodeling of neural network, leading to further exacerbation of disease behaviors[47,48]. The present study shows that microglia in the CeA are activated in mice with acute restraint stress-induced anxiety-like behaviors, accompanied by increased engulfment of GABA<sup>CeA</sup> neuronal dendritic spines and larger contact areas with these neurons. Moreover, pharmacological inhibition of microglial activity prior to acute stress does not prevent the onset of the anxiety-like behaviors but does block the relief of this phenotype in mice, suggesting that microglial activation in the CeA is responsible for the process of relieving acute stress-induced anxiety-like behaviors rather than mediating their occurrence. It follows that microglia respond to GABA<sup>CeA</sup> neuronal activation and then drive a negative feedback mechanism similar to that of inhibitory neurons, which is essential for protecting the brain from excessive activation in health and disease[49]. There are some potential limitations to examine synaptic engulfment in our study. For example, we failed to provide a direct evidence for proving the specificity of GAD65/67 antibody, although it has been widely used in immunofluorescence staining

in many studies[50,51]. In addition, the autofluorescence of lipofuscin can be likely detected within microglial lysosomes in the adult mouse brain by light microscopy[52]. To address this general issue in the field, we used the same aged mice as corresponding controls to detect microglial engulfment, which was able to minimize this effect as possible.

In addition to microglia, astrocytes, historically thought to provide structural and metabolic support for neurons, have also been implicated in synaptic pruning. However, astrocytes usually engulf and eliminate synapses during development and in conditions such as Alzheimer's disease and epilepsy[53], and there is currently no evidence supporting a role of astrocytes in synaptic pruning in response to stressful stimuli. Oligodendrocytes, another important type of glial cell, are known to be primarily responsible for the regulation of axon growth, and the production and maintenance of the myelination. Although oligodendrocyte precursor cells engulf synapses during circuit remodeling in mice, mature oligodendrocytes are not directly involved in synaptic pruning[54]. Based on the rapid response of microglia to acute stress and its characteristics of synaptic pruning, microglia probably play a dominant role in regulation of the fading of acute stress-induced anxiety-like behaviors. The function of astrocytes and oligodendrocytes under acute stress conditions needs to be further investigated.

The classic neuron–microglia interaction pathway is neuron-derived fractalkine (CX3CL1) regulation of microglial activation mediated by binding CX3CR1 on microglia[55]. However, the signaling pathways responsible for controlling neuronal secretion of CX3CL1 remain poorly understood. In the investigation of the upstream signals that mediate CX3CL1 expression, we identified the germinal center kinase subfamily III family kinase MST4, a 55-kD serine-threonine kinase highly expressed in the placenta, thymus, lymphocytes, and peripheral blood leukocytes[56,57]. MST4 is a component of the "striatal-interacting phosphatase and kinase" complex, which functions in a wide range of fundamental processes, such as signal transduction, cell cycle control, apoptosis, vesicle trafficking, and cell migration[58].

Our study provides evidence showing that MST4 is widely distributed in brain regions responsible for emotional processes. More specifically, MST4 is decreased in the CeA, and its overexpression can alleviate GABA<sup>CeA</sup> neuronal hyperactivity, reversing the anxiety-like behaviors, in ARS mice. We also found that the pro-inflammatory transcription factor NF-κB regulates Cx3cl1 transcription in GABA<sup>CeA</sup> neurons, leading to microglial activation via binding with CX3CR1. Local administration of the CX3CR1 receptor antagonist JMS-17-2 blocks the gradual extinction of anxiety-like behaviors, which occurs within 12 h post-stress induction in model mice, ultimately resulting in sustained high GABA<sup>CeA</sup> neuronal activity. Notably, MST4 overexpression can directly inhibit CX3CL1 release through a MST4-NF-κB-CX3CL1 signaling pathway that mediates neuron–microglia interactions in the early stage of acute stress to increase microglial activation and engulfment of GABA<sup>CeA</sup> neuronal spines, consequently inhibiting GABA<sup>CeA</sup> neuronal activity in ARS mice. This work provides initial evidence supporting the role of the anti-inflammatory factor MST4, specifically expressed in GABA<sup>CeA</sup> neurons, in stimulating fractalkine secretion to activate CeA microglia after acute stress.

These collective findings depict the molecular- and cellular-level neural basis underlying the extinction of anxiety-like behaviors following acute stress. This inhibitory feedback process involves acute stress-induced pro-inflammatory signaling by GABA<sup>CeA</sup> neurons to activate microglia, which, in turn, attenuate GABA<sup>CeA</sup> neuronal activity via neuronal spine engulfment to suppress anxiety-like behaviors within 12 h after stress induction. Our research thus indicates that microglial activation in the early stage of acute stress can play a protective role in regulating neuronal activity, which is different from the pro-nociceptive role of chronic stress described in previous studies.

## Methods

### Animals
All the conducted experiments were approved by the Care Committee of the University of Science and Technology of China (USTC) (UST-CACUC25120123086). Without special instruction, C57BL/6J, *Ai14* (RCL-tdT), *Cx3cr1-GFP*, and *GAD2-Cre* male mice were aged 8–10 weeks. Animals were obtained from Charles River or The Jackson Laboratory and bred at the animal facility at USTC. The mice were group-housed with five per cage in a colony in a stable environment (23–25 °C ambient temperature and 50% humidity) unless a cannula or tetrode array was implanted. They were maintained under a 12-h light–dark cycle (lights on from 7:00 to 19:00) with water and food available *ad libitum*.

### Acute restraint stress
Mice were immobilized in modified plastic syringes once for 2 h without special instruction. Holes were drilled in the ends of the syringes to allow the mice to breathe. During the restraint period, the control mice were allowed to freely move in the cage without water or food provided. Acute restraint in mice was conducted between 8:00-10:00 am, and behavioral assays were subsequently measured at 0.5 h, 4 h, 8 h, and 12 h post-stress treatment.

### Behavioral experiments
For the behavioral test, the mice were transported into the testing room 1 day prior to testing for habitation. Different batches of mice were used for open field test (OFT) and elevated plus maze (EPM) tests, in order to exclude the effects of previous test on the current test. During the testing session, the movement trajectories were recorded and subsequently analyzed offline using EthoVision XT software (Noldus). The entries into and time spent in the center of the open field or the open arms of the elevated plus maze were counted.

For OFT, an open field apparatus consisting of a square area (25 cm × 25 cm) and a marginal area (50 cm × 50 cm × 60 cm) was used to assay the anxiety. The mice were quickly placed in the central area of the test box and allowed to move freely. The animal behavior analysis software was used to record the activity of mice in the box, and the experiment time was usually 15 min. The inner wall and bottom of the test chamber were cleaned with 75% ethanol between tests to eliminate the odor effect. The number of times the mouse entered the central area and the total time spent in the central area were analyzed using animal behavior analysis software.

For EPM, the apparatus consisted of a central platform (6 × 6 cm), two closed arms (30 × 6 × 20 cm), and two opposing open arms (30 × 6 cm). The maze was 100 cm high from the ground. The mice were placed in the central area of the maze, with their heads facing open arms, and allowed to explore freely. The movement trajectories were analyzed offline, and the time spent in the open arms and the number of entries into the open arms were calculated after recording. The apparatus was cleaned between tests using 75% ethanol.

### In vivo multi-channel electrode recordings
The mice to be implanted with a tetrode array were kept alone. Three days after the mice adapted to the environment, four custom-made movable tetrode arrays were implanted into CeA. Homemade screw-driven microdrives with four implantable tetrodes were used to record simultaneously from multiple neurons. Signals were recorded after at least two weeks of recovery from the surgery, and mice were habituated to the cables connected to the electrode on their heads before recording. Subject mice were put in a cylindrical box wrapped with a copper mesh and allowed to freely move without any disturbance, and multichannel electrical signals were recorded throughout this period. Finally, the screw in the electrode holder was turned counterclockwise, and the tetrodes were gradually lowered by 40 µm after each daily recording test. Neuronal signals were amplified, filtered at a 300–5000 Hz bandwidth, and stored using NeuronStudio software. Then, data were exported to Offline Sorter 4 (Plexon) and NeuroExplorer 4 (NexTechnologies) for offline analysis. Units with a signal-to-noise ratio smaller than 2 were excluded from the analysis. Spike sorting was performed with a sorting method involving the T-Dis E-M algorithm built in Offline Sorter 4. The firing rates of sorted units were calculated using Neuroexplorer 4.

To identify GABA$^{CeA}$ neurons, blue-light pulses (470 nm, 2 ms duration, 20 Hz) were delivered at the end of each recording session at high frequency. Only laser-evoked and spontaneous spikes with highly similar waveforms (correlation coefficient > 0.9) were considered as originating from a single neuron. In subsequent experiments, we classified well-isolated units according to the typical firing pattern of light-evoked GABAergic neurons using an unsupervised clustering algorithm based on a κ-means method[59,60]. Specifically, spikes with a shorter half-spike width and half-valley width and higher firing rate were classified as putative GABAergic neurons.

### Immunohistochemistry, imaging, and image analysis
Mice were deeply anesthetized with isoflurane (about 15 s) and then quickly perfused with saline for 3 min and 4% paraformaldehyde solution for 4 min. The brain was removed and placed in 4% paraformaldehyde solution and then stored in a 4 °C refrigerator for overnight fixation. We then used 20% and 30% sucrose solution and placed the brain in a 4 °C refrigerator to dehydrate until it became isotonic. After embedding brain tissue with an embedding agent at −20 °C, the embedded brain tissue was cut into 40 µm coronal sections with a cryostat microtome system (CM1860, Leica). Brain slices were immersed in antifreeze solution and stored at −20 °C. For staining, the brain slices containing CeA were washed with PBS for 5 min three times. The brain slices were sealed in PBS containing 5% BSA and 5% Triton X-100 at room temperature for 1 h. The brain slices were then incubated in the primary antibody (Supplementary Data 2). Then the brain slices were incubated with 3% Triton X-100 and donkey serum in the primary antibody (Supplementary Data 2) at 4 °C overnight. Next, sections were washed three times with PBS for 5 min and incubated with secondary antibody (Supplementary Data 2) for 1.5 h at room temperature in a dark place. Finally, all brain slices were stained with DAPI (Cat. No. D9542, Sigma-Aldrich) dilution buffer (1:2000) for 3 min and washed twice. DAPI was used to label the nuclei, and all the sections were mounted with antifade mounting medium (Cat. No. H-1000, Vector). The slices were scanned and imaged using microscopes (Zeiss LSM880, Zeiss LSM980, Olympus SpinSR, and Olympus FV3000) to visualize the fluorescence signals.

### Three-dimensional reconstruction
The 40-µm slices were stained with anti-Iba1 for 24 h, followed by Alexa Fluor 488-conjugated secondary antibody or Alexa Fluor 643-conjugated secondary antibody staining. Imaging was performed on a Zeiss LSM880 or Zeiss LSM980 microscope using a 40×/1.3NA oil objective, and imaging parameters (laser power, gain, and offset) were consistent across all experiments. Z stacking was performed with 1.0-µm steps in the Z direction, and 512 × 512-pixel resolution images were analyzed using IMARIS 9.6.2 software (Bitplane). The IMARIS function "Fliaments" was used to quantify the microglial process length and number of branch points. Individual mice were used as independent samples.

Microglial engulfment was analyzed to create a 3D surface rendering of the microglia, with a threshold established to ensure accurate reconstruction of microglial processes, which was then used for subsequent reconstructions. The function of "Spots" was used to reconstruct the GAD65/67 puncta. Finally, the IMARIS MATLAB-based (MathWorks, Natick, MA, USA) plugin "Split into Surface Objects" was used to assess the number of GAD65/67 puncta entirely within the microglial surface (distance ≤ 0 µm). Two slices randomly picked from

each mouse were imaged and quantified for five or six mice in each group. The mean result was used for morphological analysis.

For analysis of microglial processes and neuronal dendrites contacts, imaging was performed on a Zeiss LSM880 microscope. The "Surface" function was used to accurately reconstruct both Iba1+ microglia and YFP+ dendrites, and initially established thresholds were then used for the following reconstructions. Finally, the MATLAB-based IMARIS plugin "Surface-Surface Contact Area" was used to measure the size of contact areas between microglial processes and neuronal dendrites.

## Flow cytometry

Cx3cr1-GFP mice were deeply anesthetized with pentobarbital sodium (20 mg/kg, i.p.). Subsequently, mice were perfused intracardially with 20 ml of cold Hank's Balanced Salt Solution (HBSS), followed by a rapid collection of the bilateral CeA, which was washed with cold PBS and chopped into small pieces on ice. Small tissue was mechanically homogenized using 23G needle, producing cell suspensions which were filtered through a 70 μm cell strainer. Subsequently, the samples were strained and centrifuged at 4 °C (300 $g$, 10 min). The pellets were resuspended in 38% percoll solution and centrifuged at 4 °C (800 $g$, 10 min). After careful removal of the supernatant, the pellets were rinsed with PBS. Finally, cells were measured using CytoFLEX (Beckman Coulter, USA) flow cytometer and data were analyzed by FlowJo V10. GFP+ cells were sorted by BD FACSAria III (BD, USA) for further experiments: investigation the gene expression of Tnf-α, Il-1β and Il-6 by qPCR (Applied Biosystems, ThermoFisher, China).

## Drug administration

Minocycline hydrochloride (Mino) (Cat# M9511, Sigma-Aldrich) was dissolved in water (10 mg/ml), which was administered via i.p. injection with 50 mg/kg. For intracranial microinfusion, the guide cannula (internal diameter 0.35 mm, RWD) was generally implanted 0.2 mm higher than the target nucleus of the CeA in mice two weeks before behavioral tests to ensure sufficient recovery time. A concentration of 40 mM Mino diluted in artificial cerebrospinal fluid (ACSF) was delivered using a single injection by insertion of the internal cannula, which was 0.2 mm deeper than the guide cannula, and ACSF (250 nl) was locally applied as a control.

CX3CR1 antagonist JMS-17-2 (Cat# HY-123918, MedChemExpress) was dissolved in corn oil (10 mg/ml), which was administered via i.p. injection with 10 mg/kg. A concentration of 10 mM JMS-17-2 diluted in corn oil intracranial microinfused into the bilateral CeA, and the vehicle solution (corn oil, 150 nl) was locally applied as a control.

Clozapine-N-oxide (CNO) (Cat# C0832, Sigma-Aldrich) was dissolved in DMSO (10 mg/ml) as a stock solution and diluted in 0.9% saline for DREADDs experiments, where it was injected (5 mg/kg, i.p.) 30 min before behavioral tests.

Pyrrolidinedithiocarbamate ammonium (PDTC) (Cat# HY-18738, MedChemExpress) was dissolved in PBS. A concentration of 50 mM PDTC was delivered into the bilateral CeA, and the vehicle solution (ACSF, 150 nl) was locally applied as a control.

## Stereotaxic surgery

All virus injection procedures followed the Laboratory Biosafety Guidelines approved by the USTC. Before surgery, the mice were fixed in a stereotactic frame (RWD) under an i.p. injection of pentobarbital sodium (20 mg/kg). The core temperature of the mice was maintained at 36 °C using a heating pad. A pulled glass microelectrode was backfilled with virus and connected to a 10-microliter syringe. The injection volume of different viruses varied from 100 to 300 nl depending on the viral titer and expression potential, and the infusion rate was 50 nl/min. The coordinates were included as medio-lateral (ML) from the midline (in mm), anterior–posterior (AP) from bregma, and dorso-ventral (DV) from the pial surface of the brain.

For chemogenetic manipulation, the rAAV-Ef1α-DIO-hM3D(Gq)-mCherry-WPRE-pA (AAV-DIO-hM3Dq-mCherry, AAV2/9, 1.18 × 10$^{13}$ vg/ml, Cat# PT-0042, BrainVTA) virus or rAAV-Ef1α-DIO-hM4D(Gi)-mCherry-WPRE-pA (AAV-DIO-hM4Di-mCherry, AAV2/9, 3.69 × 10$^{13}$ vg/ml, Cat# PT-0043, BrainVTA) was bilaterally injected into the CeA of GAD2-Cre mice. Three weeks after viral injection, an intraperitoneal injection of CNO (5 mg/kg, Sigma), which is a synthetic ligand for human muscarinic engineered receptors that binds and activates hM3Dq and hM4Di for the chemogenetic manipulation of neuronal activity[61], was given 30 min before the behavioral tests. The rAAV-Ef1α-DIO-mCherry-WPRE-pA (AAV-DIO-mCherry, AAV2/9, 5.31 × 10$^{12}$ vg/ml, Cat# PT-0013, BrainVTA) virus was used as the control.

To specifically label neuronal dendritic spines of GABAergic neurons, the cell type-specific sparse labeling by injection of AAV-sparse-CSSP-YFP-8E3 (AAV-CSSP-YFP-8E3, Cat# BC-SL003, BrainCase) virus into CeA in 5-week GAD2-Cre mice. AAV-CSSP-YFP-8E3 virus is produced by co-packaging the AAV-CMV-DIO-Flp plasmid and AAV-FDIO-EYFP plasmid with the ratio of 1:8E3 in a single rAAV production step (total virus titer = 5.71 × 10$^{12}$ vg/ml).

For overexpression and knockdown of MST4 in GABA$^{CeA}$ neurons, the rAAV-Ef1α-DIO-MST4-P2A-mCherry-WPRE-pA (AAV-MST4, AAV2/9, 5.97 × 10$^{12}$ vg/ml) viruses and rAAV-CMV-DIO-mCherry-U6-shRNA (MST4) (AAV-shMST4, AAV2/9, 5.26 × 10$^{12}$ vg/ml) viruses were used for knockdown and overexpression experiments, respectively. AAV-DIO-mCherry virus and rAAV-CMV-DIO-mCherry-U6-shRNA (scramble) (AAV-scramble, AAV2/9, 5.03 × 10$^{12}$ vg/ml, Cat# PT-2788, BrainVTA) virus were used as the controls. The coordinates of the injection site were confirmed in mice after Nissl dye injection. The final coordinates relative to bregma were as follows: AP, −1.2 mm; ML, ±2.80 mm; DV, −4.00 mm.

## Electrophysiological recordings

For brain slices preparation, the mice were anesthetized with isoflurane (around 15 s) followed by pentobarbital sodium (20 mg/kg, i.p.) and afterward intra-cardinally perfused with 20 mL oxygenated ice-cold N-methyl-D-glucamine (NMDG) and ACSF containing 93 NMDG, 2.5 KCl, 1.2 NaH$_2$PO$_4$, 30 NaHCO$_3$, 20 N-2-hydroxyethyl piperazine-N-2-ethane sulfonic acid (HEPES), 25 glucoses, 2 thiourea, 5 Na-ascorbate, 3 Na-pyruvate, 0.5 CaCl$_2$, 10 MgSO$_4$, and 3 glutathione (GSH) (in mM, pH: 7.3−7.4, osmolality: 300−305 mOsm). After perfusion, the brain slices (300 μm) containing CeA were quickly sectioned by chilled (2−4 °C) NMDG ACSF on a microtome (VT1200s, Leica, Germany) vibrating at 0.18 mm/s velocity. The brain slices were quickly incubated in oxygenated NMDG ACSF (saturated with 95% O$_2$/5% CO$_2$ to provide stable pH and continuous oxygenation) for 10−12 min at 33 °C and then incubated in HEPES ACSF containing 92 NaCl, 1.2 NaH$_2$PO$_4$, 2.5 KCl, 20 HEPES, 30 NaHCO$_3$, 25 glucose, 2 thiourea, 2 CaCl$_2$, 5 Naascorbate, 3 Na-pyruvate, 2 MgSO$_4$, and 3 GSH (in mM, pH: 7.3−7.4, the osmolality: 300−310 mOsm) for at least 1 h at 25 °C. Then the brain slices were transferred to a slice chamber (Warner Instruments, Holliston, MO, USA) for electrophysiological recording with continuous perfusion with standard ACSF that contained 129 NaCl, 3 KCl, 2.4 CaCl$_2$, 20 NaHCO$_3$, 10 glucose, 1.3 MgSO$_4$, and 1.2 KH$_2$PO$_4$ (in mM, pH: 7.3−7.4, osmolality: 300-310 mOsm). The slices were moved to the chamber that was constantly perfused with oxygenated standard ACSF containing 124 NaCl, 2.4 CaCl$_2$, 5 KCl, 1.3 MgSO$_4$, 26.2 NaHCO$_3$, 1.2 KH$_2$PO$_4$, and 10 glucose (in mM, pH: 7.3−7.4 osmolality: 300−305 mOsm/kg) for recording at 33 °C by solution heater (TC-344B, Warner Instruments, Holliston, MO, USA). The observer was blinded to the experimental purposes during recording and analyses.

For whole-cell patch-clamp recordings, neurons were visualized using an infrared (IR)-differential interference contrast (DIC) microscope (BX51WI, Olympus, Tokyo, Japan) with a 40× water-immersion objective. Whole-cell patch-clamp recordings were performed using

patch pipettes (5–8 MΩ) that were pulled from borosilicate glass capillaries (VitalSense Scientific Instruments Co., Ltd., Wuhan, China) with an outer diameter of 1.5 mm on a four-stage horizontal puller (P-1000, Sutter Instruments, Novato, CA, USA). The signals were collected using a MultiClamp 700B amplifier, lowpass filtered at 2.8 kHz, digitized at 10 kHz with a Digidata 1440 A Data Acquisition System, and analyzed using pClamp 10.7 (Molecular Devices, Sunnyvale, CA, USA). Neurons were excluded with a series resistance of more than 30 MΩ or that changed by more than 20% during the recording.

To record the intrinsic membrane properties, micropipettes are filled with an internal solution containing $K^+$. The intracellular solution contained (in mM): 2 $MgCl_2$, 5 KCl, 10 HEPES, 130 K-gluconate, 2 Mg-ATP, 0.6 EGTA, and 0.3 Na-GTP (pH: 7.4, osmolarity: 290–300 mOsm).

Unless otherwise stated, all drugs used for electrophysiological recording were purchased from Sigma-Aldrich, St. Louis, MO, USA.

## Western blotting

The CeA tissues were quickly obtained from 300 μm-thick slices taken on the vibratome. To extract total protein, the tissues were homogenized in ice-cold RIPA buffer, which contained 50 mM Tris–HCl (pH 7.6), 1% Triton X-100, 150 mM NaCl, 0.1% SDS, a protease inhibitor cocktail, and 0.5% sodium deoxycholate. Then, the proteins were obtained by centrifuging at 12,000 g at 4 °C for 15 min, and the protein concentrations were measured by BCA assay. In each lane, 10–20 μg protein was separated via 10% SDS-PAGE gel electrophoresis and then transferred to polyvinylidene difluoride (PVDF) membranes (Bio-Rad). After blocking with 5% skim milk, the membranes were incubated with primary antibodies, including MST4 (1:1000, Abcam, ab52491) and beta-actin (1:1000, Absin, abs137975), at 4 °C overnight and with peroxidase-labeled secondary antibody (1:5000, Jackson) at room temperature for 90 min. All the protein bands were visualized using high-sensitivity ECL reagent (GE Healthcare) and analyzed using ImageJ software. Unprocessed scans of the blots are provided in the Source Data file.

## RNA isolation and qPCR

Mice were anesthetized with isoflurane followed by pentobarbital sodium (20 mg/kg, i.p.) and intracardially perfused with ice-cold PBS. Brain slices were sectioned as for electrophysiological recordings, and CeA samples were manually dissected using a syringe needle on ice. Total RNA was isolated and purified using the RNeasy kit (Qiagen), followed by reverse transcription using a SuperRT III Reverse Transcription kit (Biosharp, BL1013B). qPCR from 1 to 10 ng of cDNA templates was performed using a universal SYBR qPCR Master Mix (Biosharp, BL697A) on an ABI StepOne system (Applied Biosystems, Waltham, MO, USA). *Gapdh* mRNA quantification was used as a loading control for normalization. Fold changes of mRNA levels over controls were calculated using the $2^{-\Delta\Delta CT}$ method. Each real-time PCR reaction was performed in triplicate. Sequences of the primers (Sangon Biotech) used in PCR are provided in Supplementary Data 3.

## Quantification and statistical analysis

The statistical analyses and graphing were performed using GraphPad Prism 8.0 (GraphPad Software, Inc., San Diego, CA, USA). Statistical significance was defined as $p < 0.05$. We performed simple statistical comparisons using Student's $t$-test, and ANOVA (one-way and two-way) was used to statistically analyze the experimental groups' data with multiple comparisons. Unpaired or paired Student's $t$-tests were used to compare the two groups. Significance levels are indicated as *$p < 0.05$; **$p < 0.01$; ***$p < 0.001$. n.s. represents not significant. All behavioral experiments except for measuring the changes in exploratory behaviors (i.e., OFT and EPM) were performed at least twice. All data are expressed as the mean ± SEM.

## Reporting summary

Further information on research design is available in the Nature Portfolio Reporting Summary linked to this article.

## Data availability

There are no restrictions on data availability in the manuscript. The data generated in this study are provided in the Source Data file. Source data are provided with this paper.

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

## Acknowledgements

This work was supported by the National Key Research and Development Program of China (STI2030-Major Projects 2021ZD0203100 to Z.Z.), the National Natural Science Foundation of China (grants 32025017 to Z.Z., 32121002 to Z.Z., 82171218 to Y.J., 82101300 to D.C., U22A20305 to Y.J.), the Plans for Major Provincial Science & Technology Projects (202303a07020002 to Z.Z.), the CAS Project for Young Scientists in Basic Research (YSBR-013 to Z.Z.), the China National

Postdoctoral Program for Innovative Talents (BX20220283 to P.C.), Youth Innovation Promotion Association CAS, CAS Collaborative Innovation Program of Hefei Science Center (2021HSC-CIP013 to Y.J.), the Fundamental Research Funds for the Central Universities (WK9100000030 to Y.J.), USTC Research Funds of the Double First-Class Initiative (YD9100002018 to Y.J.), the Natural Science Foundation of Anhui Province (2208085J30 to Y.J.), the Innovative Research Team of High-level Local Universities in Shanghai (SHSMU-ZDCX20211902 to Z.Z.), the Institute of Health and Medicine (OYZD20220007 to Z.Z.), the China Postdoctoral Science Foundation (2023M733395 to P.C.), and USTC Tang Scholar to Y.J. We thank the Confocal lmaging Unit at the Core Facility Centre for Life Science of the University of Science and Technology of China and X.S. for valuable technical expertise and assistance.

## Author contributions

P.C., Y.J., and Z.Z. conceptualized the study; D.C., Q.L., and X.S. performed most of the experiments; F.K., A.L., C.Z., and Y.L. conducted some of the behavioral experiments; D.C., P.C., Y.J., and Z.Z. were responsible for writing; D.W. and S.Q. were involved in the revision of the final manuscript; D.C., P.C., Y.J., and Z.Z. supervised the study.

## Competing interests

The authors declare no competing interests.
