## [Peer Review File · Nature Communications]

REVIEWER COMMENTS

Reviewer #1 (Remarks to the Author):

The study by Chen et al has provided multiple lines of evidence showing that the extinction of acute stress-induced anxiety is controlled by microglia activation, which is caused by elevated CeAGABA neuronal CX3CL1 secretion via MST4-NFκB signaling. Activated microglia in turn inhibit CeAGABA neuronal activity via the engulfment of dendritic spines, leading to the extinction of anxiety-like behaviors induced by restraint stress. These results have provided an interesting mechanism on neuron-microglia interactions, which may contribute to the maintenance of brain homeostasis. While technical strengths are impressive, a few major concerns on experimental procedures raise concerns on the validity of results and interpretations.

1. Timeline of events. Fig. 1, anxiety and CeAGABA neuronal activity are increased at 8hrs post-ARS, but not 12hrs post-ARS. What triggers the dramatic changes within just 4 hrs? CX3CL1 secretion, microglia activation, and engulfment of dendritic spines of CeAGABA neurons all happened within the last 4 hrs?

2. Behavioral assays. Subsequent EPM tests usually significantly affect (reduce) the animals' "anxiety" measurements (time and entries in open arms) of prior EPM tests. Repeated EPM tests at close intervals (0.5h, 4h, 8h, 12h) of the same animals are highly problematic (Fig. 1d, e). The heatmaps (Fig. 1d) show no or little time on open-arm at 0.5-8hr post-ARS, and a striking reversal at 12hr post-ARS, which is not very representative of the statistics (Fig. 1e). Repeated OFT tests may have similar problems.

3. Timing of tests. Many comparison experiments were carried out with a 12hr interval (0.5h vs. 12h post-ARS), which means that these measurements were conducted at the time points that mice have significantly different levels of activity. This could significantly affect the behavioral, in vivo electrophysiological and maybe even immunocytochemical results.

4. Intracranial microinfusion. Fig. 2 used cannular implantation to mouse CeA for minocycline (microglial inhibitor) injection. Guide cannula usually causes a large lesion of the implanted area, which could trigger microglia activation. Although ACSF injection is used as a control here, the effect of minocycline could be due to microglia activation by lesion rather than 2hr restraint stress. When was the multichannel electrode implanted? Was CeA examined after cannular injection for tissue integrity?

5. The increase of CX3CL1 secretion at 0.5hr post-ARS (Fig. 4b) is not compelling. It is also puzzling how CX3CL1 mRNA could be increased by 2hr ARS (Fig. 4a).

6. The data with JMS-17-2 (CX3CR1 antagonist) used i.p. injections for microglia and spine measurements and local (cannular) injections for behavioral and electrophysiological experiments, why were the different routes of injections used? When was the multichannel electrode implanted?

7. Fig. 5. The justification of examining the participation of MST4 is very weak (line 260-262). NFκB can be regulated by many molecules other than MST4. The involvement of NFκB lacks solid evidence. To more convincingly demonstrate the role of MST4 in anxiety extinction, the level of MST4 at 12h post ARS, as well as the effect of MST4 on CX3CL1 secretion, needs to be examined.

Reviewer #2 (Remarks to the Author):

Chen et al, authors of the manuscript, "Microglia govern extinction of acute stress-induced anxiety" present data that indicate neuron-microglia interactions are important mediators of behavioral responses following acute restraint stress. This is an interesting paper with several cutting-edge approaches used to demonstrate microglia may contribute to observed neurobiological and behavioral outcomes. Despite this, there are concerns about data interpretation and limitations of experimental approaches detract from the impact of this work. This work is likely to be of interest to the broad readership of Nature Communications, but the points outlined below should be addressed prior to publication.

Main points:

1) The initial data demonstrate that stress-induced activation of CeA GABA neurons is related to and sufficient to modulate behavioral responses to acute restraint stress. This raises the question as to why the authors wanted to add microglia, and not other non-neuronal cell types, into this model. Specifically, why are microglia uniquely suited to regulate this response? Perhaps this can be discussed.

2) One major concern is the interpretation of behavioral tests, in this case the OPT and EPM. Traditionally these tests were considered measures of 'anxiety-like behavior'. However, it is accepted now that rodent models do not recapitulate the complexity of psychiatric disorders, such as anxiety disorders. The behavioral outcomes reflect domains relevant to psychiatric disorders but it is not entirely clear how this relates to clinical cases. It is recommended that the authors limit use of 'anxiety-like behavior'. It would be more suitable to report this as changes in exploratory behavior or aversion to

novel environments. Either way the authors should report the specific outcomes (i.e., decreased time in center or open arms) and then describe the potential significance in the Conclusion.

3) Related to behavioral testing it is unclear if mice were repeatedly tested in Fig.1b-e. This is important as rodents will adapt their responses after exposure to a novel environment.

4) Density and morphological features of microglia is not sufficient to determine their functional state. Just based on immunohistology results it is unclear if these cellular responses are related to increased neuroinflammation or an alternate phenotype. It is recommended that further molecular or cellular characterization be performed.

5) It is intriguing that microglia density in the BLA particularly, fluctuates over such short time frames. Prior studies indicate that microglia turnover and proliferation is low (in the absence of injury). The authors should validate these changes in density with markers for proliferation or cell death.

6) Minocycline should not be considered an inhibitor of microglia. Since it was administered centrally it is likely influences molecular and cellular pathways in multiple cell types. As such, the authors should temper their conclusions regarding this approach.

7) In Fig.3, it appears that dendritic segments were used as individual samples. This is not appropriate, because dendritic segments from one mouse should not be considered independent samples. Segments from each mouse should be averaged to generate a cumulative average and then statistical analyses should be carried out on these samples. As is, the sample size artificially increases the statistical power and over-estimates group differences.

8) There are other concerns about the immunohistology in Fig.3. First, the CD68 immunolabeling seems particularly intense at the 0.5 h timepoint. Enlarged images should be presented to validate co-localization with IBA1. Second, immunohistology and 3D image analyses are used to suggest that microglia are engulfing GABA neuron structures (GAD65/67). The authors have rendered the synaptic markers and other puncta (i.e., colored 'nodes'), and it is recommended that these manipulations be removed. Moreover, these results are questionable as it appears that even in control mice there is an unusually high number of inclusions in microglia. This is further exaggerated in mice exposed to acute restraint. The authors need to reassess the specificity of their antibodies and their thresholds for image analyses. Also any engulfed synaptic structures should be within lysosomes, so the authors should validate these inclusions with CD68 immunolabeling. Finally, it is not clear how this data was quantified and 'normalized' in Fig.3I. Again individual cells should be considered independent samples.

9) The connection between stress-induced neuronal activity in the BLA and CX3CL1 signaling is not apparent. There are several neuroimmune signaling pathways altered by changes in neuronal activity. More rationale and supporting data for focusing on CX3CL1 should be provided. Also CX3CL1 is a chemokine, not a pro-inflammatory cytokine, which should be corrected throughout the manuscript.

10) Beyond the rationale for targeting CX3CL1 there is an issue with interpretation of results in Fig.4g-l. Significant differences are not reported for vehicle controls at 0.5 and 12 hours on all the outcomes. This limits interpretation and does not support the conclusion that targeting CX3CL1 signaling with JM-17-2 and microglia are involved in the observed neurobiological, behavioral, or neurophysiological effects.

11) Related to the point above, it is unclear how MST4 was connected to CX3CL1 signaling in the brain. It is recommended that primary data in extended figures showing MST4 localization in neurons be included in the main figures. This is important as it provides direct evidence that targeted molecules are expressed in cells of interest. As described MST4 is an important regulator of NF- κ B signaling. In this context, you would expect that it would influence other cytokines and chemokines. It is recommended that other molecular targets including IL-1b, IL-6, and TNFa be examined.

12) Several figures lack comprehensive statistical analyses. As noted, some important group differences are not reported and this limits data interpretation.

Other points:

- Studies used only male mice. This should be emphasized in the Results and Discussion.

- Sample sizes (as in the # of mice used) should be reported in figure legends.

- The summary figure in Extended Data is simplified and better suited for a review manuscript. It is recommended that it be removed.

Reviewer #3 (Remarks to the Author):

In this manuscript, Chen et al. investigates a novel role of microglia. The immediate behavioral outcome of an acute stress is a somewhat understudied element of the stress response. Chen et al. highlight the role of central amygdala inhibitory neurons in controlling anxiety-like behaviors following stress. They propose that microglia play a crucial role in the behavioral recovery via the engulfment of dendritic spines. The authors reveal the pathway necessary for the activation of microglia by inhibitory CeA neurons.

Overall, I find the manuscript very interesting and novel and was particularly impressed with the rigor with which the experiments seemed to be conducted and analyzed. The concerns I have are primarily related to the terminology and interpretation of the behavioral results.

1) Traditionally the term 'extinction' is a learning process where the repeated exposure to a cue without reinforcement/punishment leads to the fading of a behavior. In the manuscript, the behavioral analysis is not based on cue triggered behaviors nor repeated exposures. I believe the consistent use of a different term describing the fading or disappearance of the behavioral state evoked by acute stress would be very beneficial.

2) The authors should show the distance data collected during open field or elevated plus maze exposures to make the claim that the behaviors reported are indeed anxiety-like and not just the results of decreased locomotion.

3) The identification of inhibitory neurons during multi-channel recordings is challenging even with optogenetical tagging. The authors claim in the results section that in vivo multielectrode recordings showed an increase in the activity of CeA inhibitory. How can they be sure if in the methods section only 'putative CeA GABA' is mentioned? What is the reference for the identification?

Typo: throughout the paper it says 'Extended Date Fig.' instead of Extended Data Fig.

**Response to referees**

**Manuscript ID:** NCOMMS-23-10335A

**Title:** Microglia govern the fading of acute stress-induced anxiety

We sincerely appreciate the positive and helpful evaluation from the editor and Reviewers. In
light of these thoughtful critiques, we have performed additional experiments to address the
specific concerns. We have also substantially revised the manuscript and incorporated these
suggestions and comments into the revised manuscript. The revised version of our study with
tracked changes (highlighted in blue) has been uploaded as a separate file. Detailed changes
and our point-by-point responses to Reviewers' questions are presented below.

**Contents:**

Response to Reviewer 1: Page 2-14

Response to Reviewer 2: Page 15-27

Response to Reviewer 3: Page 28-31

**Reviewers' Comments:**

**Reviewer #1:**

The study by Chen et al has provided multiple lines of evidence showing that the extinction of
acute stress-induced anxiety is controlled by microglia activation, which is caused by elevated
CeA^{GABA} neuronal CX3CL1 secretion via MST4-NFκB signaling. Activated microglia in turn
inhibit CeA^{GABA} neuronal activity via the engulfment of dendritic spines, leading to the
extinction of anxiety-like behaviors induced by restraint stress. These results have provided an
interesting mechanism on neuron-microglia interactions, which may contribute to the
maintenance of brain homeostasis. While technical strengths are impressive, a few major
concerns on experimental procedures raise concerns on the validity of results and
interpretations.

1. Timeline of events. Fig. 1, anxiety and CeA^{GABA} neuronal activity are increased at 8hrs post-
ARS, but not 12hrs post-ARS. What triggers the dramatic changes within just 4 hrs? CX3CL1
secretion, microglia activation, and engulfment of dendritic spines of CeA^{GABA} neurons all
happened within the last 4 hrs?

**Response:** First, we would like to thank the Reviewer for their positive review of our work and
insightful guidance towards improving our study.

To address this concern, we conducted new experiments to characterize the timeline of
dynamic changes in CX3CL1 secretion, microglial activity, and dendritic spines of CeA^{GABA}
neurons at 8 h post treatment in ARS-2h mice. Our findings in the original manuscript have
shown that dendritic spine number of CeA^{GABA} neurons significantly increases at 0.5 h post-
ARS-2h, suggesting a higher rate of spine formation than elimination rate. By contrast, we now
found that the number of dendritic spines at 8 h post ARS-2h treatment was less than that at 0.5
41 h post treatment, suggesting that the rate of dendritic spine formation decelerated with time post
stress induction (please see Response Document Figure 1a, b, and also see new Figure 3b, c).
These results indicate that the process of spine formation at progressively slower rates
continued until the microglia engulfment activity returned to normal levels, consistent with
results that showed dendritic spine number at 12 h post ARS-2h was not significantly different
from that in the control group (please see original Figure 2b, c and 3b, c).

We also found that CX3CL1 expression and microglial activity in the CeA were
significantly higher than that in control mice at 8 h post ARS-2h, but were slightly lower at this
time point compared to 0.5 h post ARS-2h (please see Response Document Figure 1c-f, and
also see new Figure 2a-d, Figure 4a, b). These results were consistent with the observed changes
in spontaneous spike firing rate of CeA^{GABA} neurons, which together indicated that CX3CL1
secretion, microglial activation, and engulfment of CeA^{GABA} neuronal dendritic spines were
already occurring after ARS-2h, and gradually decreased over time until returning to control

levels by 12 h post ARS-2h, that is, not just within the last 4 hours from 8-12 h post treatment.
 Collectively, these results indicate that the activation of microglia in the CeA, and associated
 changes in their engulfment activity, can gradually recover to control levels with increasing
 time after restraint-induced stress, providing further insights into the mechanisms underlying
 anxiety-like behaviors.

These new results are now included in the revised manuscript.

**Response Document Figure 1. Changes in CeA^{GABA} neuronal dendritic spines, CX3CL1**
 **secretion, and microglial activity following ARS-2h.**

(a, b) Representative images of neuronal dendrites (a) and summarized data for spine numbers
 65 per 10 μm (b) (n = 6 mice per group). Scale bars, 10 μm.

(c, d) Representative images (c) and quantitative analyses (d) of Western blot analysis of
 soluble CX3CL1 expression in CeA tissues from control, 0.5 h, and 8 h post treatment ARS-2h
 mice (n = 5 mice per group).

(e) Representative images of Iba1 immunostaining and 3D reconstruction of microglia in the
 CeA of ARS-2h mice at 0.5 h/8 h post-stress induction and corresponding control mice. Scale
 bars, 40 μm (overview) and 20 μm (inset and rendering).

(f) Quantification of Iba1⁺ cell numbers and Imaris-based semi-automatic quantification of cell
 morphometry, including total process length and number of branch points of Iba1⁺ microglia in
 the CeA at 0.5 h/8 h post-treatment in ARS-2h and control mice (n = 6 mice per group).

All data are presented as mean ± SEM. **p* < 0.05, ***p* < 0.01, and ****p* < 0.001. See also Table
 S1.

2. Behavioral assays. Subsequent EPM tests usually significantly affect (reduce) the animals'
 "anxiety" measurements (time and entries in open arms) of prior EPM tests. Repeated EPM

tests at close intervals (0.5h, 4h, 8h, 12h) of the same animals are highly problematic (Fig. 1d,
e). The heatmaps (Fig. 1d) show no or little time on open-arm at 0.5-8hr post-ARS, and a
striking reversal at 12hr post-ARS, which is not very representative of the statistics (Fig. 1e).
Repeated OFT tests may have similar problems.

**Response:** Thanks for raising this question. We completely agree with the Reviewer's
comments and we are grateful for the opportunity to correct an unintentional oversight in the
figure legends. Actually, in our original manuscript, different batches of mice were used for
each anxiety-related behavioral assay, including OFT and EPM tests in order to avoid the very
impacts of previous tests that the Reviewer mentioned. Therefore, as a result, the number of
animals used for behavioral experiments differs among time points in the original submitted
manuscript (please see original Table S1).

In addition, the apparent discrepancy between the heatmap and statistics of the OFT and
EPM tests are due to random selection and variability among individual mice in behavioral
performance. As suggested, we have replaced all non-representative heatmaps for the OFT and
EPM tests, and provided more information necessary to clarify our experimental procedures in
the revised manuscript.

3. Timing of tests. Many comparison experiments were carried out with a 12hr interval (0.5h
vs. 12h post-ARS), which means that these measurements were conducted at the time points
that mice have significantly different levels of activity. This could significantly affect the
behavioral, in vivo electrophysiological and maybe even immunocytochemical results.

**Response:** We completely agree with the Reviewer's comments, and again, we regret any
confusion caused by our omission of some important details about the behavioral experiments.

There is indeed a wide range of state- and time-dependent activities in organisms, such as
the classic circadian rhythms¹. Consistently, after reanalysing OFT data, we also found that the
control mice travelled significantly greater distances at 12 h (~10:00 pm) than at 0.5 h (~10:30
am) (please see Response Document Figure 2, and also see new Supplementary Figure 1),
which is consistent previous studies that established mice are nocturnal and therefore exhibit
more activity at night^{2,3,4,5}. In fact, all ARS-2h treatments in our original manuscript were
conducted between 8:00-10:00 am, and anxiety-like behaviors were subsequently measured at
different times. Notably, we found no significant difference in distance travelled in OFT tests
at 0.5 h/4 h/8 h/12 h post ARS-2h from that of corresponding control mice (Response Document
Figure 2, and also see new Supplementary Figure 1).

We agree that independent mice should be used for this experiment when repeated at
different time points, with corresponding controls for each time point, to exclude time-
dependent effects. Although we followed this approach during the experiment, we only used
immunohistochemistry data from the control group at 0.5 h post-ARS to compare with other

groups in order to provide a clear and simplified data presentation. To address concerns raised
 by the Reviewer, we now include all control group data collected in 12 hours following ARS-
 2h, and comparisons if immunohistochemistry data between treatment and control groups are
 performed with samples from the same time point avoid potential time-dependent effects on
 our conclusions. More specifically, no differences in microglial activation, dendritic spines of
 CeA^{GABA} neurons, expression levels of the phagocytic marker, CD68, or microglial
 phagocytosis were detected between control and ARS-2h mice at 12 h post treatment (please
 see Response Document Figure 3, and also see new Figure 2a-d, Figure 3a-c, 3g-l and
 Supplementary Figure 9). Furthermore, no differences were detected between control mice at
 0.5 h post-ARS-2h and control mice at 12 h post-ARS-2h (please see Response Document
 Figure 3, and also see new Figure 2a-d, Figure 3a-c, 3g-l and Supplementary Figure 9). Notably,
 we also did not find differences between 0.5 h and 12 h post-ARS-2h control group in the
 neuronal activity detected by *in vivo* multichannel recordings (please see original Figure 1h).

In the revised manuscript, we now compare the control groups between the first and last
 time points to show that there is no difference between them (please see new Figure 1h). These
 results are now presented in the revised manuscript.

**Response Document Figure 2. Total distance of ARS-treated mice in OFT test at different**
 **time points.**

Summarized data of movement distances in central area of the OFT test in ARS-2h mice at 0.5
 138 h, 4 h, 8 h, and 12 h post-stress induction and corresponding control mice. Different batches of
 139 mice were used for each OFT assay (0.5 h, n = 9 mice per group; 4 h, n = 9 mice per group; 8
 140 h, n = 11 mice per group; 12 h, n = 8 mice per group).

All data are presented as mean ± SEM. **p* < 0.05; n.s., not significant. See also Table S1.

Response Document Figure 3. The changes in microglial activation, dendritic spines of CeA^{GABA} neurons, expression levels of phagocytic marker CD68, and microglial engulfment following ARS.

(a) Representative images of Iba1 immunostaining and 3D reconstruction of microglia in the CeA of ARS-2h mice at 0.5 h/4 h/8 h/12 h post-stress induction and corresponding control mice. Scale bars, 40 µm (overview) and 20 µm (inset and rendering).

(b) Quantification of Iba1⁺ cell numbers and Imaris-based semi-automatic quantification of cell morphometry, including the total process length and number of branch points of Iba1⁺ microglia in the CeA at 0.5 h/12 h post-stress induction in ARS-2h and control mice (n = 6 mice per group).

(c) Representative images of neuronal dendrites (top) and summarized data for spine numbers per 10 µm (bottom) (n = 6 mice per group). Scale bars, 10 µm.

(d) Representative images (left) and quantitative analyses (right) of immunostaining for CD68 (red), Iba1 (green), and DAPI (blue) in the CeA from corresponding control and 0.5 h/12 h post-stress induction mice (n = 6 mice per group). Scale bars, 20 µm.

(e) Reconstructed images (left) and summarized data (right) of microglia-dendrite contact size between Iba1⁺ microglia (red) and YFP⁺ neuronal dendrites of GABAergic neurons in the CeA of control or 0.5 h/12 h post-ARS-2h mice (0.5 h control, n = 128 cells from six mice; 0.5 h post-ARS-2h, n = 126 cells from six mice; 12 h control, n = 124 cells from six mice; 12 h post-

ARS-2h, n = 123 cells from six mice). Scale bars, 10 μm (overview) and 5 μm (inset and
rendering).

(f) Reconstructed images (left) and summarized data (right) for the number of microglia-
dendritic spines of Iba1⁺ microglia (red) containing YFP⁺ neuronal dendritic spines of
GABAergic neurons in the CeA from corresponding control or 0.5 h/12 h post-stress induction
mice (n = 6 mice per group). Scale bars, 5 μm .

(g) Representative images and 3D surface rendering of Iba1⁺ microglia (green) containing
GAD65/67⁺ puncta (red) and DAPI (blue) (left), and Quantification of GAD65/67⁺ puncta in
microglia (right) in the CeA from corresponding control and 0.5 h/12 h post-ARS-2h mice (n
= 53 cells from six mice per group). Scale bars, 10 μm (overview) and 5 μm (inset and
rendering).

All data are presented as mean \pm SEM. ** $p < 0.01$, and *** $p < 0.001$; n.s., not significant. See
also Table S1.

4. Intracranial microinfusion. Fig. 2 used cannular implantation to mouse CeA for minocycline
(microglial inhibitor) injection. Guide cannula usually causes a large lesion of the implanted
area, which could trigger microglia activation. Although ACSF injection is used as a control
here, the effect of minocycline could be due to microglia activation by lesion rather than 2hr
restraint stress. When was the multichannel electrode implanted? Was CeA examined after
cannular injection for tissue integrity?

**Response:** We thank the Reviewer for pointing out this unintentional omission of details in the
original manuscript that has resulted in confusion. In fact, according to methods described in
previous studies^{6,7}, to more effectively deliver the drug, the guide cannula was generally placed
0.2 mm higher than the target nucleus of the CeA. Actually, to minimize damage during drug
delivery, treatments were administered in a single injection by insertion of the internal cannula,
which should be 0.2 mm deeper than the guide cannula. In addition, we specifically avoided
repeated insertion of projection dummy cannula to ensure that we did not cause repeated
activation of glial cells. Moreover, after each behavioral test, we checked the accuracy of
cannular placement and the integrity of the CeA, and excluded data from mice that had a
misplaced cannula or damaged brain tissues. As for the implanted time, the cannula and
multichannel electrode were implanted in mice two weeks before behavioral tests to ensure
sufficient recovery time, as shown in Figure 2d of the originally submitted manuscript.

However, there is still no doubt that guide cannula causes a large lesion in the implanted
area. To address the Reviewer's concerns, we now provide imaging data showing the cannular
implantation sites and performed experiments examining gliosis in the CeA. The results showed
that the integrity of CeA brain tissue remained intact (*i.e.*, undamaged) after cannular
implantation by the method described above (please see Response Document Figure 4, and also

see new Supplementary Figure 7). Furthermore, immunofluorescent staining for gliosis
 markers, Iba1 and glial fibrillary acidic protein (GFAP), in brain slices of mice with implanted
 cannula showed that although fluorescence signal of Iba1⁺ microglia and GFAP⁺ astrocytes
 were significantly more abundant in the brain area (Region A, above the CeA) reached by the
 cannula tip than in the corresponding brain area of control mice, no difference was found in the
 fluorescence signal of these glial cells within the CeA between controls and mice with
 implanted cannula (Region B) (please see Response Document Figure 4, and also see new
 Supplementary Figure 7). These results suggest that, in our experiments, there was no damage
 to the target brain area of the CeA due to this strict cannular implantation.

These findings thus indicate that activation of microglia in the CeA was indeed caused by
 ARS-2h, not cannular implantation, in the current study. These new results have been
 incorporated into the revised manuscript.

**Response Document Figure 4. Immunofluorescent staining of Iba1 and GFAP in the CeA**
 **and adjacent regions of naive mice with or without cannular implantation.**

(a, b) Representative image (a) and enlarged image (b) of immunostaining for Iba1 and GFAP
 around the cannula position in the CeA of implanted mice and the same position in control mice.
 Scale bars, 100 μm (a) and 20 μm (b).

(c) Quantitative analyses of immunostaining for Iba1 and GFAP in (b) (n = 6 mice per group).

All data are presented as mean \pm SEM. *** $p < 0.001$; n.s., not significant. See also Table S1.

5. The increase of CX3CL1 secretion at 0.5hr post-ARS (Fig. 4b) is not compelling. It is also
puzzling how CX3CL1 mRNA could be increased by 2hr ARS (Fig. 4a).

**Response:** These concerns are worth discussing and we have conducted additional experiments
to address this issue. We agree with the Reviewer that careful examination of the increase in
CX3CL1 protein levels in CeA tissues at 0.5 h post treatment in the ARS-2h group compared
to control group in original Figure 4b is indeed not very convincing, and this change is even
less obvious in the gel bands. Further scrutiny of the data from Actin gel bands shows that total
protein content is inconsistent between samples, with higher total protein content in the control
group than in the 0.5 h post-treatment samples of the ARS-2h group. As a result, the gel itself
appears to show no difference in CX3CL1 between groups, while the quantitative image
analysis shows significant differences after adjusting for this variability. To address this issue,
we repeated WB experiments examining changes in CX3CL1 protein level. After ensuring
consistency in the loading of Actin control protein on the gel, we found that CX3CL1 protein
expression was higher in ARS-2h samples collected at 0.5 h post treatment compared to the
corresponding control group (please see Response Document Figure 5, and also see new Figure
4a, b).

Regarding the Reviewer's concern about "how to increase CX3CL1 mRNA in 2-hr ARS",
this result is indeed puzzling. Previous animal studies have shown that stress could increase
neuroinflammation levels in the brain^{8,9}, including elevated expression of NF- κ B and
CX3CL1^{10,11,12}. In the original manuscript, we observed that ARS-2h induced an increase in *Nf-*
*κ b* transcript levels, consistent with other published data¹¹. Since *Cx3cl1* gene expression is
regulated by the transcription factor, NF- κ B, the elevation in *Cx3cl1* mRNA levels after 2 hours
of acute stress is reasonable. In addition, our new experimental results showed a significant
decrease in *Cx3cl1* mRNA expression was observed at 0.5 h post-ARS-2h compared to that in
vehicle control animals, following PDTC, a selective inhibitor of NF- κ B, pre-administrated into
the CeA (please see Response Document Figure 6d, and also see new Figure 4p). These findings
provide evidence that the NF- κ B signalling pathway is involved in regulating CX3CL1 protein
expression in acute stress states.

We present these results and an accompanying description in the revised manuscript.

**Response Document Figure 5. The expression of CX3CL1 protein at 0.5 h post ARS-2h.**

**(a, b)** Representative gel images **(a)** and quantitative analyses **(b)** of Western blot detection of
 soluble CX3CL1 protein in CeA samples from control and ARS-2h mice at 0.5 h post-stress
 induction (n = 5 mice per group).

All data are presented as mean ± SEM. ***p* < 0.01. See also Table S1.

6. The data with JMS-17-2 (CX3CR1 antagonist) used i.p. injections for microglia and spine
 measurements and local (cannular) injections for behavioral and electrophysiological
 experiments, why were the different routes of injections used? When was the multichannel
 electrode implanted?

**Response:** We regret any confusion due to missing details or vague descriptions of our results
 in the originally submitted manuscript.

In this study, all *in vivo* pharmacological experiments were administered via implanted
 cannula except for multichannel electrophysiological recordings, which required drug
 administration via intraperitoneal (i.p.) injection. We acknowledge that local administration can
 indeed better demonstrate the function of the target brain area, but due to technological and size
 limitations of the CeA, it is currently not feasible to implant both a multichannel electrode and
 a cannula, simultaneously, in the same brain area of mice. For this reason, we used a different
 strategy for drug injection for experiments requiring multichannel electrodes.

In addition, although i.p. administration of JMS-17-2 may affect a wider range of brain
 areas, *in vivo* electrophysiological recording data in the original manuscript showed that
 CeA^{GABA} neurons in model mice still exhibited high-frequency discharge levels at 12 h post-
 ARS-2h. At a minimum, these results suggested that increased activity of the CX3CL1
 signalling pathway in CeA^{GABA} neurons is indeed involved in the extinction of anxiety-like
 behaviors in mice with restraint-induced acute stress, and at least partially supported the
 reliability of results obtained by *in vivo* multichannel electrophysiological recordings in mice

with JMS-17-2 injections (i.p.).

In this study, the multichannel electrodes were implanted in mice two weeks before i.p. drug
administration to ensure sufficient recovery time for the mice^{13,14}. The restraint treatment was
applied for 2 hours after drug administration, and the signal was recorded at relevant time points.
As mentioned in comment #4 by this Reviewer, we now provide details of the multichannel
electrode implantation process in the revised manuscript.

7. Fig. 5. The justification of examining the participation of MST4 is very weak (line 260-262).
NFκB can be regulated by many molecules other than MST4. The involvement of NFκB lacks
solid evidence. To more convincingly demonstrate the role of MST4 in anxiety extinction, the
level of MST4 at 12h post ARS, as well as the effect of MST4 on CX3CL1 secretion, needs to
be examined.

**Response:** We greatly appreciate this astute and highly constructive comment. As suggested,
we conducted additional experiments to address this issue and extended our discussion of this
topic in the revised manuscript.

Regarding of the Reviewer's concern that "The involvement of NF-κB lacks solid
evidence", we sought to establish a firm, experimentally well-supported relationship between
NF-κB and CX3CL1 following acute stress induction. To this end, we administrated
Pyrrolidinedithiocarbamate ammonium (PDTC), a selective inhibitor of NF-κB, via implanted
cannula in the CeA. In these mice, the extinction of ARS-2h-induced anxiety-like behaviors
was blocked following PDTC administration, accompanied by a significant decrease in *Cx3cll*
mRNA expression compared to that in control animals (please see Response Document Figure
6, and also see new Figure 4m-p). These findings suggested that the NF-κB signalling pathway
is involved in regulating CX3CL1 expression and extinction of anxiety-like behaviors.

As recommended, we also sought to demonstrate the role of MST4 in anxiety extinction
through new experiments examining MST4 protein levels in the CeA at 12 h after ARS-2h, as
well as the effect of MST4 on CX3CL1 secretion. Our results showed that MST4 protein levels
returned to that of controls at 12 h post-ARS-2h (please see Response Document Figure 7, and
also see new Figure 5c), while MST4 knockdown resulted in a significant increase in CX3CL1
protein accumulation in the CeA of naïve (non-ARS) mice compared levels in control mice
without MST4 knockdown (please see Response Document Figure 8, also see new Figure 6e).
These results were consistent with the increased *Cx3cll* mRNA expression in the ARS-2h mice
that we reported in the originally submitted manuscript (Extended Data Fig. 10c). Taken
together, these findings support the hypothesis that MST4-NF-κB play a functional role in the
extinction of anxiety-like behaviors by modulating CX3CL1 levels in the CeA.

In addition, we note that little background is provided for MST4 in the Introduction of the
original manuscript. Previous reports have shown that acute stress can lead to increased

microglial activation and heightened inflammatory response in the brain, and has even been
 associated with activation of the CX3CL1/CX3CR1 signaling pathway in the amygdala and
 hippocampus^{12,15,16,17}. This activation of inflammatory response can be detrimental to neuronal
 function and may disrupt homeostatic balance in the central nervous system. During innate
 immune response, Toll-like receptors recognize pathogen-associated molecular patterns, and
 rapidly activate the immune inflammatory response through the signaling molecule, TRAF6
 (TNF receptor associated factor 6), to facilitate the elimination of pathogens^{18,19,20,21}. In healthy
 organism, the inflammatory immune response requires precise regulation to mediate pathogen
 clearance without also damaging the host, that is, immune homeostasis. Mammalian sterile20-
 like kinase 4 (MST4) has been previously reported to directly phosphorylate TRAF6, inhibiting
 its ubiquitination, and subsequently limiting inflammatory response²². Thus, MST4 could act
 as a “brake” on TLR-TRAF6-mediated inflammatory responses. By contrast, TRAF6 is known
 to play a pivotal role in NF- κ B activation and TLR4 (toll like receptor 4) pathway-mediated
 macroautophagy/autophagy activation^{21,23}. Following the induction of acute stress, the MST4-
 NF- κ B-CX3CL1 signaling pathway exerts an important role in maintaining immune
 homeostasis during inflammatory response.

This background for MST4 has been added to the revised manuscript, along with the above
 new results and accompanying text.

**Response Document Figure 6. Inhibition of NF- κ B signaling in the CeA blocks the**
 **extinction of acute stress-associated anxiety-like behaviors.**

**(a)** Experimental scheme for pretreatment of ARS-2h mice with vehicle or PDTC.

**(b)** Summarized data for number of entries and time spent in central area in OFT by ARS-2h

mice pre-treated with vehicle or PDTC (0.5 h: ARS-2h + Vehicle, n = 8 mice, ARS-2h + JMS-

17-2, n = 8 mice; 12 h: ARS-2h + Vehicle, n = 8 mice, ARS-2h + JMS-17-2, n = 8 mice).
 (c) Summarized data for number of entries and time spent in open arms of EPM by ARS-2h
 mice pre-treated with vehicle or PDTC (0.5 h: ARS-2h + Vehicle, n = 8 mice, ARS-2h + JMS-
 17-2, n = 8 mice; 12 h: ARS-2h + Vehicle, n = 8 mice, ARS-2h + JMS-17-2, n = 8 mice).
 (d) qPCR analysis of *Cx3cl1* mRNA expression in the CeA of ARS-2h mice pre-treated with
 vehicle or PDTC (n = 5 mice per group).
 All data are presented as mean ± SEM. **p* < 0.05, ***p* < 0.01, and ****p* < 0.001; n.s., not
 significant. See also Table S1.

 **Response Document Figure 7. MST4 protein levels return to that of control mice at 12 h**
 **after ARS-2h.**
 (a, b) Representative gel images (a) and quantitative analyses (b) for Western blot detection of
 MST4 protein in CeA tissues from control and ARS-2h mice at 12 h post-stress induction (n =
 5 mice per group).
 All data are presented as mean ± SEM. n.s., not significant. See also Table S1.

 **Response Document Figure 8. MST4 knockdown in naïve mice increases CX3CL1 protein**
 **accumulation in the CeA.**

(a) Schematic for bilateral virus infection into the CeA of *GAD2-Cre* mice.
(b) Representative gel images for Western blot detection of soluble CX3CL1 expression in the
CeA of naïve mice infected with AAV-mCherry or AAV-shMST4-mCherry.
(c) Quantitative analyses for CX3CL1 bands for the gel in (b) (n = 5 mice per group).
All data are presented as mean ± SEM. ***p* < 0.01. See also Table S1.
We would like to take this opportunity to again thank the Reviewer for their very helpful
guidance and highly constructive questions about our study.

**Reviewer #2:**

Chen et al, authors of the manuscript, "Microglia govern extinction of acute stress-induced
anxiety" present data that indicate neuron-microglia interactions are important mediators of
behavioral responses following acute restraint stress. This is an interesting paper will several
cutting-edge approaches used to demonstrate microglia may contribute to observed
neurobiological and behavioral outcomes. Despite this, there are concerns about data
interpretation and limitations of experimental approaches detract from the impact of this work.
This work is likely to be of interest to the broad readership of Nature Communications, but the
points outlined below should be addressed prior to publication.

**Main points:**

1. The initial data demonstrate that stress-induced activation of CeA^{GABA} neurons is related to
and sufficient to modulate behavioral responses to acute restraint stress. This raises the question
as to why the authors wanted to add microglia, and not other non-neuronal cell types, into this
model. Specifically, why are microglia uniquely suited to regulate this response? Perhaps this
can be discussed.

**Response:** We would first like to thank the Reviewer for their supportive comments and
guidance in improving our study.

In this manuscript, we found that ARS causes an increase in the activity of CeA^{GABA}
neurons, resulting in anxiety-like behaviors within 12 hours; at the same time, the number of
CeA^{GABA} neuronal dendritic spines also increases in ARS-2h mice, then gradually decreases
over time until returning to control levels by 12 h post ARS-2h. Based on these findings, we
speculated that synaptic pruning could potentially mediate this change in CeA^{GABA} neuronal
plasticity.

In the central nervous system, two main types of glial cells are involved in synaptic
pruning, including microglia and astrocytes^{24,25,26}. Microglia regulate synaptic pruning, a
critical process in refining neural circuits, through various signaling pathways: First, microglia
express complement system proteins, such as C1q and C3, which tag synapses for microglial
engulfment^{27,28,29,30,31}; Second, microglia can interact with neurons through signaling molecules
that are regulated by neuronal activity, such as fractalkine (CX3CL1) and its receptor
(CX3CR1), to modulate synaptic pruning^{25,32,33}; Third, microglia have been shown to bind IL-
33 released from astrocytes via IL1RL1, activating their function in neuronal synaptic pruning
in the nerve injury mice model^{34,35,36}.

Astrocytes were historically thought to provide structural and metabolic support for
neurons³⁷. More recent studies have shown that astrocytes can directly engulf and eliminate
synapses during development and in conditions such as Alzheimer's disease and epilepsy^{28,38}.
In addition, astrocytes can also contribute to synaptic pruning by releasing factors that guide

microglia to specific synapses targeted for elimination^{25,38}. However, there is currently no
evidence supporting a role of astrocytes in synaptic pruning in response to stressful stimuli.

Oligodendrocytes, another important type of glial cell, are primarily responsible for the
regulation of axon growth, and the production and maintenance of the myelination in the central
nervous system. Some limited evidence suggests that mature oligodendrocytes are not directly
involved in synaptic pruning, although oligodendrocyte precursor cells engulf synapses during
circuit remodeling in mice³⁹. Based on our evidence obtained while establishing our current
ARS-2h murine model, together with the reasons described above, we focused on microglia
rather than astrocytes and oligodendrocytes in the current study.

We have added some text related to this topic to the revised Discussion section.

2. One major concern is the interpretation of behavioral tests, in this case the OPT and EPM.
Traditionally these tests were considered measures of ‘anxiety-like behavior’. However, it is
accepted now that rodent models do not recapitulate the complexity of psychiatric disorders,
such as anxiety disorders. The behavioral outcomes reflect domains relevant to psychiatric
disorders but it is not entirely clear how this relates to clinical cases. It is recommended that the
authors limit use of ‘anxiety-like behavior’. It would be more suitable to report this as changes
in exploratory behavior or aversion to novel environments. Either way the authors should report
the specific outcomes (i.e., decreased time in center or open arms) and then describe the
potential significance in the Conclusion.

**Response:** We appreciate the Reviewer’s perspective and guidance. We completely agree that
using the term “anxiety-like behavior” may be an inaccurate description of the phenotype
defined by EPM and OFT behavioral assays, since these tests cannot fully capture the
complexity of anxiety disorders^{40,41,42}. As suggested, we have replaced more than half of the
“anxiety-like behaviors” with “low level of exploratory behaviors” in the revised manuscript.

As recommended, we now refer to the observed behavioral changes in mice as alterations
in exploratory behaviors in the revised manuscript. We have also modified the paper to report
specific outcomes, such as decreased time spent in the center of OFT or open arms of EPM, to
provide a more accurate description of the observed behavioral changes, while reserving the
broader potential significance of these behavioral changes for our conclusions.

3. Related to behavioral testing it is unclear if mice were repeatedly tested in Fig.1b-e. This is
important as rodents will adapt their responses after exposure to a novel environment.

**Response:** We completely agree Reviewer’s concern that “rodents will adapt their responses
after exposure to a novel environment” in behavioral assays. Some confusion about our
experimental design has arisen due to an unfortunate oversight in our description of the
experimental details. In the originally submitted manuscript, different batches of mice from the

ARS-2h and control groups were in fact used for each exploratory behavioral assay, including
EPM and OFT tests, in order to exclude the effects of previous tests on the current test. For
these reasons, the number of animals used in behavioral experiments at each time point is not
consistent in the original version of the paper (please see original Table S1). More information
necessary to understand our experimental design has been added to the figure legends of the
revised manuscript.

4. Density and morphological features of microglia is not sufficient to determine their
functional state. Just based on immunohistology results it is unclear if these cellular responses
are related to increased neuroinflammation or an alternate phenotype. It is recommended that
further molecular or cellular characterization be performed.

**Response:** We thank the Reviewer for this valuable suggestion. Although our study reveals that
activated microglia inhibit CeA^{GABA} neuronal activity via engulfment of their dendritic spines,
ultimately leading to the fading of restraint stress-induced anxiety-like behaviors in male mice,
it is indeed unknown whether neuroinflammation is required for this phagocytosis. Previous
studies have shown that microglia are often accompanied by neuroinflammation when they are
activated and phagocytosed^{43,44,45,46}. In order to more rigorously characterize the functional state
of microglia, we conducted additional experiments, as recommended by the Reviewer,
including immunofluorescent staining for the inflammatory marker MHCII as well as a panel
of classical inflammatory factors that co-localize with Iba1. We found that MHCII levels were
significantly increased in ARS-2h mice compared with controls at 0.5 h post-stress induction
(please see Response Document Figure 9a, b, and also see new Supplementary Figure 5a, b),
which was consistent with the observed changes in microglial Iba1 expression. In addition, the
increased levels of the phagocytosis marker, CD68, in ARS-2h mice at 0.5 h post-stress
induction that showed in the original data, further indicating that phagocytic function was
enhanced in microglia of ARS-2h mice (please see original Figure 3g).

To further test whether ARS can cause classical inflammatory changes in microglia, we
used immunofluorescent staining to measure changes in TNF α , IL-1 β , and IL-6 in microglia.
The results revealed that TNF α , IL-1 β and IL-6 were all significantly upregulated in microglia
of the CeA from ARS-2h mice at 0.5 h post-stress induction compared with controls (please see
Response Document Figure 9c, and also see new Supplementary Figure 5c-h). These findings
are consistent with previous studies and support the notion that the observed cellular responses
of microglia are indeed associated with increased neuroinflammation under ARS.

These additional data are presented in the revised manuscript.

Response Document Figure 9. Immunofluorescent staining for the inflammatory marker MHCII and a panel of classical inflammatory factors in microglia.

(a, b) Representative images (a) and quantitative analyses (b) of immunostaining for MHCII
(red), Iba1 (green), and DAPI (blue) in the CeA of 0.5 h/12 h post ARS-2h and corresponding
control mice (n = 6 mice per group). Scale bars, 20 μm.

(c-h) Immunofluorescent staining (c, e, g) and quantitative analyses (d, f, h) of TNFα, IL-1β,
and IL-6 co-localized with Iba1 in the CeA of ARS-2h and control mice at 0.5 h post-treatment.
Scale bars, 20 μm.

All data are presented as mean ± SEM. **p* < 0.05, ***p* < 0.01, and ****p* < 0.001; n.s., not
significant. See also Table S1.

5. It is intriguing that microglia density in the BLA particularly, fluctuates over such short time
frames. Prior studies indicate that microglia turnover and proliferation is low (in the absence of
injury). The authors should validate these changes in density with markers for proliferation or
cell death.

**Response:** We appreciate your insightful comment about our observations of significant
fluctuation in microglia density in the CeA within the relatively short experimental timeframe.
While we agree that some prior studies show that microglial turnover and proliferation is low
in the absence of injury^{47,48,49,50}, other previous studies have shown that certain brain regions,
including the amygdala, medial prefrontal cortex, anterior cingulate cortex and hippocampus,

can rapidly activate microglia in response to stress stimuli^{12,17,51,52,53,54}. Thus, the increased
 number of microglia we detected under acute stress stimuli may be due to excessive
 proliferation and differentiation. As suggested, we carried out additional experiments using
 Ki67, a marker for proliferation, and TUNEL (cell death) assays at different time points
 following ARS-2h treatment to further validate our observations of changes in microglial
 density in ARS model mice. We found that Ki67 expression significantly increased in microglia
 in the CeA of ARS-2h mice at 0.5 h post-treatment compared to that in controls, followed by a
 gradual decrease over time (please see Response Document Figure 10a, b, and also see new
 Figure 2e-g). In contrast, TUNEL assays showed that apoptosis levels peaked at 8 hours post-
 ARS-2h, then returned to baseline levels at 12 hours post-ARS-2h treatment (please see
 Response Document Figure 10c, d, and also see new Supplementary Figure 6). These findings
 further illustrate the dynamic activation of microglia and subsequent restoration to resting levels
 in the CeA following ARS-2h treatment.

We present these results in the revised manuscript.

**Response Document Figure 10. Immunofluorescent staining for Ki67 and TUNEL assays**
 **following ARS-2h treatment in mice.**

**(a, b)** Representative images **(a)** and quantitative analyses **(b)** of immunostaining for Ki67 (red),
 Iba1 (green), and DAPI (blue) in the CeA of ARS-2h and control mice at 0.5 h post-stress
 induction (n = 6 mice per group). Scale bars, 10 μ m.

**(c, d)** Representative images **(c)** and quantitative analyses **(d)** of TUNEL assays; fragmented
 DNA (red), Iba1 (green), and DAPI (blue) in the CeA of ARS-2h and control mice at 0.5 h/8

519 h/12 h post-treatment (n = 6 mice per group). Scale bars, 10 μ m.

All data are presented as mean \pm SEM. * p < 0.05, and *** p < 0.001; n.s., not significant. See
also Table S1.

6. Minocycline should not be considered an inhibitor of microglia. Since it was administered
centrally it is likely influences molecular and cellular pathways in multiple cell types. As such,
the authors should temper their conclusions regarding this approach.

**Response:** We understand the Reviewer's concerns and appreciate their insight. In the original
manuscript, we describe minocycline as a "microglial inhibitor... used for selective
pharmacological inhibition of microglia", which we now understand to be inappropriate. We
have replaced that description with "minocycline, which has been widely used to inhibit
microglial activity," in the revised manuscript. As suggested, we also appropriately temper our
conclusions regarding the specific effects of minocycline on microglia in the revised paper.

7. In Fig.3, it appears that dendritic segments were used as individual samples. This is not
appropriate, because dendritic segments from one mouse should not be considered independent
samples. Segments from each mouse should be averaged to generate a cumulative average and
then statistical analyses should be carried out on these samples. As is, the sample size artificially
increases the statistical power and over-estimates group differences.

**Response:** Thanks for this helpful guidance. We completely agree with the Reviewer's
suggestion. However, we found that, regrettably, some details of the data analysis were left out
of the original manuscript. Actually, each data point represents one cell, not one dendritic
segment, in the plots of spine density in our original manuscript (please see original Table S1).
In fact, at least 3–5 dendritic segments were averaged for each analyzed cell, and about twenty-
five cells from six total mice were examined in each group. As suggested, we have reorganized
our data to clearly show that each symbol represents one mouse in the revised manuscript
(please see new Figure 3c, 3f, Figure 4h, and also see new Supplementary Figure 9d).

8. There are other concerns about the immunohistology in Fig.3. First, the CD68
immunolabeling seems particularly intense at the 0.5 h timepoint. Enlarged images should be
presented to validate co-localization with IBA1. Second, immunohistology and 3D image
analyses are used to suggest that microglia are engulfing GABA neuron structures (GAD65/67).
The authors have rendered the synaptic markers and other puncta (i.e., colored 'nodes'), and it
is recommended that these manipulations be removed. Moreover, these results are questionable
as it appears that even in control mice there is an unusually high number of inclusions in
microglia. This is further exaggerated in mice exposed to acute restraint. The authors need to
reassess the specificity of their antibodies and their thresholds for image analyses. Also any

engulfed synaptic structures should be within lysosomes, so the authors should validate these
inclusions with CD68 immunolabeling. Finally, it is not clear how this data was quantified and
‘normalized’ in Fig.3l. Again individual cells should be considered independent samples.

**Response:** Thanks for the careful examination of our figures and this helpful guidance. As
suggested, we have added the relevant enlarged images to validate co-localization of CD68
with Iba1 in mice at 0.5 h and 12 h post-ARS-2h treatment (please see Response Document
Figure 11a, b, and also see new Figure 3g). In addition, we have removed the colored puncta
from images in original fig. 3k, which are actually the neuronal nuclei stained by DAPI (please
see Response Document Figure 11c, d, and also see new Figure 3h).

Regarding the Reviewer’s concern that “these results are questionable as it appears that
even in control mice there is an unusually high number of inclusions in microglia”, we believe
that there may be two reasons for this. First, we checked the related studies of the GAD65/67
antibody (ab183999, abcam) used in this study. The WB experiments of the antibody are
provided by the official website of Abcam ([https://www.abcam.cn/products/primary-
antibodies/gad65--gad67-antibody-epr19366-ab183999.html](https://www.abcam.cn/products/primary-antibodies/gad65--gad67-antibody-epr19366-ab183999.html)), which showed two bands with
antibodies against GAD67 and GAD65 fragment recombinant proteins in mouse. In addition,
the GAD65/67 antibody have been widely used in immunofluorescent staining in numerous
studies^{55,56}. Second, although there are many GAD65/67 punctas in microglia from control mice,
we found that the macrophage marker CD68 is rarely expressed in the control mice (please see
original Figure 3g, h), so there should not be as much microglial engulfment in the control mice.
Therefore, it seems that the low threshold adjustment in the algorithm of the microglial
engulfment analysis results in unusually high number of inclusions in microglia from control
mice. Briefly, the Imaris MATLAB-based (MathWorks) plugin “Split into Surface Objects” was
used to assess the number of GAD65/67 puncta in microglia (distance $\leq 0 \mu\text{m}$). Based on this
analytical method, the data of the engulfed synaptic marker are from the GAD65/67 puncta that
entirely within microglia as well as those distributed on these cell surface. As suggested, we
have readjusted the “Estimated XY Diameter” from $0.8 \mu\text{m}$ to $0.9 \mu\text{m}$, which is used to estimate
the size of GAD65/67 puncta. After adjusting the parameters, we found that the number of
GAD65/67 puncta in microglia from the control groups and ARS-2h mice were significantly
reduced, but the increase in the number of GAD65/67 puncta in CeA microglia at 0.5 h post-
ARS-2h compared with control mice, was still remained (please see Response Document
Figure 11c, d, and also see new Figure 3h). These results indicate that the threshold is indeed
too low.

Again, we regret any confusion stemming from insufficient detail about our methods. We
have added the information necessary to fully understand these experiments in the revised
Methods section.

To address the suggestion that inclusions require validation by CD68 immunolabeling, we

have conducted additional experiments in which microglia are co-labeled with CD68 and
GAD65/67. Confocal imaging data and 3D surface rendering further depict abundant
colocalization between immunoreactive puncta of GAD65/67, CD68, and Iba1⁺ microglia in
the CeA of ARS-2h mice at 0.5 h, but not at 12 h post-treatment, nor in the corresponding
control animals (please see Response Document Figure 11e, f, and also see new Figure 3i).
These results are fully consistent with our previous findings that show an increase in
colocalization of GAD65/67 immunoreactive puncta with Iba1⁺ microglial processes in the CeA
at 0.5 h post-ARS-2h, compared with control mice, and that this phenomenon no longer occurs
at 12 h post-ARS-2h (please see original Figure 3k, l). These results thus confirm that engulfed
synaptic puncta are present within microglia in the CeA of ARS-2h mice.

Regarding the concern that “how data were normalized and quantified in Figures 3l”, in
fact, mice were randomly selected from the control and ARS groups to collect phagocytosis
data from intact microglia. In addition, to normalize the data, we used the mean value obtained
from the control group as a reference and expressed data from the treatment group as a
percentage of this mean. Furthermore, as mentioned above and following the Reviewer’s
recommendation, individual cells were used as independent samples.

These results are all presented in the revised version of the paper.

**Response Document Figure 11. Microglial engulfment of synaptic structures in ARS-2h**
 **mice at 0.5 h/12 h post-stress induction.**

(a, b) Representative images (a) and quantitative analyses (b) of immunostaining for CD68
 (red), Iba1 (green), and DAPI (blue) in the CeA of ARS-2h mice and corresponding controls at
 0.5 h/12 h post-stress induction (n = 6 mice per group). Scale bars, 20 μ m.

(c) Representative images and 3D surface rendering of Iba1⁺ microglia (green) containing
 GAD65/67⁺ puncta (red) in the CeA of ARS-2h and control mice at 0.5 h/12 h post-stress
 induction. Scale bars, 50 μ m (overview) and 10 μ m (inset and rendering).

(d) Quantification of GAD65/67⁺ puncta in microglia of mice from (c) (n = 53 cells from six
 mice per group).

(e, f) Representative images (e) and quantitative analyses (f) of immunostaining for CD68 (red),

Iba1 (green), and GAD65/67⁺ puncta (purple) in the CeA of ARS-2h and control mice at 0.5
623 h/12 h post-stress induction (n = 24 cells from six mice per group). Scale bars, 5 μm.

All data are presented as mean ± SEM. ****p* < 0.001; n.s., not significant. See also Table S1.

9. The connection between stress-induced neuronal activity in the BLA and CX3CL1 signaling
is not apparent. There are several neuroimmune signaling pathways altered by changes in
neuronal activity. More rationale and supporting data for focusing on CX3CL1 should be
provided. Also, CX3CL1 is a chemokine, not a pro-inflammatory cytokine, which should be
corrected throughout the manuscript.

**Response:** We are grateful for this very interesting and important question that could provide
additional mechanistic insight into how microglia govern the fading of acute stress-induced
anxiety.

To address the comment that “There are several neuroimmune signaling pathways altered
by changes in neuronal activity”, we used qPCR to screen for expression of a number of
molecules previously reported to be involved in the response to stress and to mediate neuronal-
microglia interactions^{57,58,59,60,61,62}, including inflammatory chemokines (CX3CL1, CCL2,
CXCL10), inflammatory cytokines (TNFα, IL-1β, IL-6, IL-4, IL-10, IL-33, IL-34, TGFβ,
CSF1), complement proteins (C1q, C3), and growth factor (BDNF). We found that the mRNA
levels of *Cx3cl1*, *Ccl2*, *Il-1β*, *Tnfα*, and *Il-6* (please see Response Document Figure 12, and also
see new Supplementary Figure 12) were all significantly higher in the CeA of ARS-2h mice
than in control animals at 0.5 h.

It is well known that the CX3CL1 is a secreted chemokine specifically expressed in
neurons and engaged in microglia-neuron interactions that is widely reported to be involved in
activity-dependent synaptic pruning of neurons^{25,53}. While, CCL2, one of the most potent
microglia/macrophage chemokines, is predominantly produced by astrocytes and resident
microglia, and to a lesser extent, by endothelial cells^{63,64,65}, and there are few studies on the
involvement of CCL2 in synaptic pruning. In addition, TNFα, IL-1β, IL-6 are mainly released
from microglia and our new results have shown that the increased levels of *Tnfα*, *Il-1β*, and *Il-6*
mRNA are consistent with the upregulation of these cytokines in CeA microglia at 0.5 h post
ARS-2h (please see Response Document Figure 9c-h, and also see new Supplementary Figure
5c-h). It should be mentioned that activation of CeA^{GABA} neurons precedes microglial activation
in the CeA of ARS mice (please see original Extended Data Fig. 5), we therefore focused our
study on the neuron-specific release of CX3CL1, which has been previously shown to play a
role in emotional responses and trigger microglial engulfment in the brain in response to
stressful stimuli⁵³.

Finally, we thank the Reviewer for pointing out our misnomer of CX3CL1 as a pro-
inflammatory cytokine. We have carefully checked the manuscript and used the correct term

throughout the revised manuscript. We provide these new results and expanded descriptions of
our methods in the revised manuscript.

**Response Document Figure 12. Expression of factors related to neuroimmune signaling**
**pathways in the CeA of ARS-2h mice.** qPCR analysis of mRNA levels of cytokines,
chemokines, complement proteins, and growth factors in the CeA of ARS-2h mice (n = 5 mice
666 per group).

All data are presented as mean ± SEM. * $p < 0.05$, ** $p < 0.01$, and *** $p < 0.001$. See also Table
S1.

10. Beyond the rationale for targeting CX3CL1 there is an issue with interpretation of results
in Fig.4g-l. Significant differences are not reported for vehicle controls at 0.5 and 12 hours on
all the outcomes. This limits interpretation and does not support the conclusion that targeting
CX3CL1 signaling with JM-17-2 and microglia are involved in the observed neurobiological,
behavioral, or neurophysiological effects.

**Response:** Thanks for this helpful advice. As suggested by the Reviewer, we compared data
from the vehicle controls obtained at 0.5 h and 12 h post ARS-2h. Examination of CeA^{GABA}
neuron dendritic spines revealed that spine density was obviously greater in these neurons at
0.5 h post ARS-2h than that at 12 h post ARS-2h in the vehicle control animals (please see new
Figure 4h). Furthermore, data from behavioral tests showed that vehicle control mice made
fewer entries and spent less time in the central area of the OFT and open arms of the EPM at
0.5 h post ARS-2h compared to vehicle control mice at 12 h post ARS-2h (please see new
Figure 4i, j). Electrophysiological recordings showed that CeA^{GABA} neuronal spontaneous firing
activity was also higher in vehicle control mice at 0.5 h post ARS-2h compared with that at 12
684 h post-ARS-2h (please see new Figure 4l), which was consistent with the results of behavioral
tests. Based on the evidence showing that administration of JM-17-2 could prevent the recovery
of processes by 12 h post ARS-2h observed in the vehicle control mice, it is reasonable to
conclude that the CX3CL1 signaling pathway is indeed involved in the fading of ARS-induced
reduction in exploratory behaviors, or so-called anxiety-like behaviors, in mice.

These additional supporting results have been added to the revised manuscript.

11. Related to the point above, it is unclear how MST4 was connected to CX3CL1 signaling in
the brain. It is recommended that primary data in extended figures showing MST4 localization
in neurons be included in the main figures. This is important as it provides direct evidence that
targeted molecules are expressed in cells of interest. As described MST4 is an important
regulator of NF- κ B signaling. In this context, you would expect that it would influence other
cytokines and chemokines. It is recommended that other molecular targets including IL-1b, IL-
6, and TNF α be examined.

**Response:** Thanks for the very constructive comment. As suggested, we moved Supplementary
Figure S7d to main Figure 5a and expanded our description of these results in the revised
manuscript (please see new Figure 5a).

Regarding of the Reviewer's suggestions that "It is recommended that other molecular
targets including IL-1b, IL-6, and TNF α be examined", as in response to your comment #9, we
have used qPCR to screen for the expression of 15 molecules previously reported as involved
in neuroimmune signalling pathways at the transcriptional level, including the levels of *Il-1 β* ,
*Il-6*, and *Tnfa* mRNA in the CeA at 0.5 h post treatment in ARS-2h mice. We found that
compared with control mice, the expression level of *Il-1 β* , *Il-6*, and *Tnfa* mRNA was increased
significantly in the CeA of ARS-2h mice at 0.5 h post treatment (please see Response Document
Figure 12, and also see new Supplementary Figure 12). Based on these findings, we have
provided justification and evidence for why CX3CL1 was selected instead of other
inflammatory-associated molecules for study.

This understanding, combined with our finding of decreased expression of the anti-
inflammatory protein, MST4, in CeA^{GABA} neurons after ARS (please see original Figure 5a, b),
led to our hypothesis that MST4-regulated CX3CL1 participates in synaptic pruning of
CeA^{GABA} neurons by microglia, consequently mediating the fading of stress-induced aversion
to novel environments and exploratory behaviors.

These new results are presented in the revised manuscript.

12. Several figures lack comprehensive statistical analyses. As noted, some important group
differences are not reported and this limits data interpretation.

**Response:** Thanks for this helpful guidance. We have provided information regarding statistical
tests and data to each figure legend, and as recommended, we have added comparisons of data
from the control groups at different time points post-treatment. In addition, the statistical
analysis for each figure is included in Supplementary Table S1 of the original manuscript.

**Other points:**

1. Studies used only male mice. This should be emphasized in the Results and Discussion.

**Response:** We appreciate this advice. We have updated the Results and Discussion sections of

our revised manuscript to emphasize that only male mice were used in our studies.

2. Sample sizes (as in the # of mice used) should be reported in figure legends.

**Response: Done.**

3. The summary figure in Extended Data is simplified and better suited for a review manuscript.

It is recommended that it be removed.

**Response: Done.**

The authors wish to take this opportunity to again thank the Reviewer for their careful review

of our paper and for their extremely helpful guidance that has helped us to greatly improve the

coherence and quality of our study.

**Reviewer #3:**

In this manuscript, Chen et al. investigates a novel role of microglia. The immediate behavioral
outcome of an acute stress is a somewhat understudied element of the stress response. Chen et
al. highlight the role of central amygdala inhibitory neurons in controlling anxiety-like
behaviors following stress. They propose that microglia play a crucial role in the behavioral
recovery via the engulfment of dendritic spines. The authors reveal the pathway necessary for
the activation of microglia by inhibitory CeA neurons.

Overall, I find the manuscript very interesting and novel and was particularly impressed
with the rigor with which the experiments seemed to be conducted and analyzed. The concerns
I have are primarily related to the terminology and interpretation of the behavioral results.

1. Traditionally the term ‘extinction’ is a learning process where the repeated exposure to a cue
without reinforcement/punishment leads to the fading of a behavior. In the manuscript, the
behavioral analysis is not based on cue triggered behaviors nor repeated exposures. I believe
the consistent use of a different term describing the fading or disappearance of the behavioral
state evoked by acute stress would be very beneficial.

**Response:** We would first like to thank the Reviewer for their careful examination of our text,
their supportive comments, and helpful guidance about how to improve our study. We agree
with the Reviewer’s professional comments. Indeed, the term “extinction” is most commonly
used in psychology-related studies, such as studies examining memory and fear^{66,67}. In those
experimental contexts, extinction refers to the process through which a learned behaviour or
response is eliminated through withdrawal or rewards. This elimination involves repeated
presentation of the conditioned stimulus in the absence of an unconditioned stimulus, leading
to a gradual decrease and eventual extinction of the conditioned response. Thus, fading, may
be a more appropriate term than extinction to describe the loss of a behavioral state evoked by
stress. We have revised the manuscript accordingly.

2. The authors should show the distance data collected during open field or elevated plus maze
exposures to make the claim that the behaviors reported are indeed anxiety-like and not just the
results of decreased locomotion.

**Response:** We appreciate this advice. As suggested, we now provide the data for distance
travelled by mice in the OFT. It should be noted that these data show no difference between the
experimental and control groups, suggesting that ARS-2h does not affect locomotor ability of
mice (please see Response Document Figure 14, and also see new Supplementary Figure 1),
which are consistent with previous studies^{68,69}. Additionally, we would like to mention that
different batches of mice were used for each anxiety-related behavioral assay, including EPM
and OFT, in order to avoid the impacts of the previous test on the current test. These new results

and related descriptions are now presented in the revised manuscript.

**Response Document Figure 14. Performance of ARS-2h-treated mice in open field tests at**
**different time points.**

Summarized data of distance travelled in the central area of the OFT by ARS-2h mice at 0.5 h,
4 h, 8 h, and 12 h post-stress induction and corresponding control mice. Different batches of
mice were used for each OFT assay (0.5 h, n = 9 mice per group; 4 h, n = 9 mice per group; 8
784 h, n = 11 mice per group; 12 h, n = 8 mice per group).

All data are presented as mean ± SEM. * $p < 0.05$; n.s., not significant. See also Table S1.

3. The identification of inhibitory neurons during multi-channel recordings is challenging even
with optogenetical tagging. The authors claim in the results section that in vivo multielectrode
recordings showed an increase in the activity of CeA inhibitory. How can they be sure if in the
methods section only 'putative CeA^{GABA}' is mentioned? What is the reference for the
identification?

**Response:** This issue really deserves our full attention in current multi-channel recording
experiments. We apologize for our oversight in failing to explicitly describe the identification
of inhibitory neurons in the CeA in the Methods section. Numerous previous studies have
reported that the majority of neurons in the CeA are inhibitory GABAergic neurons, which are
involved in the regulation of various emotional disorders, including anxiety and
depression^{7,70,71,72}. Although it was a previously widely held view that spikes with a shorter half-
spike width and half-valley width and higher firing rate in multi-channel electrophysiological
recordings can be classified as putative GABAergic neurons, increasing evidence suggests this
approach can be unreliable.

To provide further support for our conclusions, we conducted additional optogenetic
tagging experiments to label CeA^{GABA} neurons by a combination of optogenetic techniques and
multi-channel electrophysiological recordings in *GAD2-Cre* mice with CeA injection of AAV-
DIO-ChR2-mCherry virus. Three weeks later, optrodes were implanted at the same site where
the virus was injected. The optrode was constructed by surrounding an optical fiber (200 μm

core, Newdoon) with four tetrodes, the tips of which were 200 μm longer than the fiber^{13,14}. To
 identify CeA^{GABA} neurons, blue-light pulses (470 nm, 2 ms duration, 20 Hz) were delivered at
 the end of each recording session at high frequency. Only laser-evoked and spontaneous spikes
 with highly similar waveforms (correlation coefficient > 0.9) were considered as originating
 from a single neuron. In subsequent experiments, we classified well-isolated units according to
 the typical firing pattern of light-evoked GABAergic neurons using an unsupervised clustering
 algorithm based on a κ -means method^{73,74}. Specifically, spikes with a shorter half-spike width
 and half-valley width and higher firing rate were classified as putative GABAergic neurons in
 the CeA. These results, along with a detailed description, have been included in the revised
 manuscript (please see Response Document Figure 15, and also see new Supplementary Figure
 2).

We now present these new results and provide more information necessary to understand
 our process for neuronal classification in the revised Methods section.

 **Response Document Figure 15. Identification of characteristic CeA^{GABA} neuronal spike**
 **waveforms.**

(a) Schematic for optogenetic tagging and electrophysiological recording. Enlarged area shows
 optrodes.

(b) Representative images of virus injection site in the CeA (left) and mCherry⁺ neurons
 colocalized with immunofluorescent signal for GABAergic neurons (right). Scale bars, 50 μm
 (left) and 20 μm (right).

(c, d) Example recording of spontaneous and light-evoked spikes from a CeA^{GABA} neuron (c)
 and overlay of averaged spontaneous (red) and light-evoked (blue) spike waveforms from the
 example unit (d).

Typo: throughout the paper it says 'Extended Date Fig.' instead of Extended Data Fig.

**Response:** Done.

We again thank the Reviewer for their supportive comments and very helpful critique which
has helped us to improve our experimental rigour and ultimately increased the purport of our
conclusions.

**References**

- 1. Ayyar VS, Sukumaran S. Circadian rhythms: influence on physiology, pharmacology,
and therapeutic interventions. *J Pharmacokinet Pharmacodyn* **48**, 321-338 (2021).
- 2. Xie L, *et al.* Cholecystokinin neurons in mouse suprachiasmatic nucleus regulate the
robustness of circadian clock. *Neuron* **111**, 2201-2217 e2204 (2023).
- 3. Weinert D, Freyberg S, Touitou Y, Djeridane Y, Waterhouse JM. The phasing of
circadian rhythms in mice kept under normal or short photoperiods. *Physiol Behav* **84**,
791-798 (2005).
- 4. Kakefuda K, *et al.* Diacylglycerol kinase beta knockout mice exhibit lithium-sensitive
behavioral abnormalities. *PLoS One* **5**, e13447 (2010).
- 5. Ip JY, *et al.* Gomafu lncRNA knockout mice exhibit mild hyperactivity with enhanced
responsiveness to the psychostimulant methamphetamine. *Sci Rep* **6**, 27204 (2016).
- 6. Harvey K, *et al.* Intracranial Cannula Implantation for Serial Locoregional Chimeric
Antigen Receptor (CAR) T Cell Infusions in Mice. *J Vis Exp*, (2023).
- 7. Zhou W, *et al.* A neural circuit for comorbid depressive symptoms in chronic pain. *Nat*
*Neurosci* **22**, 1649-1658 (2019).
- 8. Sugama S, Takenouchi T, Hashimoto M, Ohata H, Takenaka Y, Kakinuma Y. Stress-
induced microglial activation occurs through beta-adrenergic receptor: noradrenaline
as a key neurotransmitter in microglial activation. *J Neuroinflammation* **16**, 266 (2019).
- 9. Yamanishi K, *et al.* Acute stress induces severe neural inflammation and overactivation
of glucocorticoid signaling in interleukin-18-deficient mice. *Transl Psychiatry* **12**, 404
(2022).
- 10. Koo JW, Russo SJ, Ferguson D, Nestler EJ, Duman RS. Nuclear factor-kappaB is a
critical mediator of stress-impaired neurogenesis and depressive behavior. *Proc Natl*
*Acad Sci U S A* **107**, 2669-2674 (2010).
- 11. Zhang Y, *et al.* Restraint stress induces lymphocyte reduction through p53 and
PI3K/NF-kappaB pathways. *J Neuroimmunol* **200**, 71-76 (2008).
- 12. Bollinger JL, Bergeon Burns CM, Wellman CL. Differential effects of stress on
microglial cell activation in male and female medial prefrontal cortex. *Brain Behav*
*Immun* **52**, 88-97 (2016).
- 13. Zhu X, *et al.* Distinct thalamocortical circuits underlie allodynia induced by tissue
injury and by depression-like states. *Nat Neurosci* **24**, 542-553 (2021).
- 14. Zhou W, *et al.* Sound induces analgesia through corticothalamic circuits. *Science* **377**,
198-204 (2022).
- 15. Bollinger JL, Collins KE, Patel R, Wellman CL. Behavioral stress alters corticolimbic
microglia in a sex- and brain region-specific manner. *PLoS One* **12**, e0187631 (2017).
- 16. Winkler Z, Kuti D, Ferenczi S, Gulyas K, Polyak A, Kovacs KJ. Impaired microglia
fractalkine signaling affects stress reaction and coping style in mice. *Behav Brain Res*
**334**, 119-128 (2017).
- 17. Nie X, *et al.* The Innate Immune Receptors TLR2/4 Mediate Repeated Social Defeat
Stress-Induced Social Avoidance through Prefrontal Microglial Activation. *Neuron* **99**,
464-479 e467 (2018).
- 18. Aderem A, Ulevitch RJ. Toll-like receptors in the induction of the innate immune
response. *Nature* **406**, 782-787 (2000).

- 19. Medzhitov R. Recognition of microorganisms and activation of the immune response.
*Nature* **449**, 819-826 (2007).
- 20. O'Shea JJ, Murray PJ. Cytokine signaling modules in inflammatory responses.
*Immunity* **28**, 477-487 (2008).
- 21. Yang FM, Chang HM, Yeh ETH. Regulation of TLR4 signaling through the
TRAF6/sNASP axis by reversible phosphorylation mediated by CK2 and PP4. *Proc*
*Natl Acad Sci U S A* **118**, (2021).
- 22. Jiao S, *et al.* The kinase MST4 limits inflammatory responses through direct
phosphorylation of the adaptor TRAF6. *Nat Immunol* **16**, 246-257 (2015).
- 23. Fang J, *et al.* TRAF6 Mediates Basal Activation of NF-kappaB Necessary for
Hematopoietic Stem Cell Homeostasis. *Cell Rep* **22**, 1250-1262 (2018).
- 24. Jessen KR. Glial cells. *Int J Biochem Cell Biol* **36**, 1861-1867 (2004).
- 25. Faust TE, Gunner G, Schafer DP. Mechanisms governing activity-dependent synaptic
pruning in the developing mammalian CNS. *Nat Rev Neurosci* **22**, 657-673 (2021).
- 26. Neniskyte U, Gross CT. Errant gardeners: glial-cell-dependent synaptic pruning and
neurodevelopmental disorders. *Nat Rev Neurosci* **18**, 658-670 (2017).
- 27. Presumey J, Bialas AR, Carroll MC. Complement System in Neural Synapse
Elimination in Development and Disease. *Adv Immunol* **135**, 53-79 (2017).
- 28. Dejanovic B, *et al.* Complement C1q-dependent excitatory and inhibitory synapse
elimination by astrocytes and microglia in Alzheimer's disease mouse models. *Nat*
*Aging* **2**, 837-850 (2022).
- 29. Hong S, *et al.* Complement and microglia mediate early synapse loss in Alzheimer
mouse models. *Science* **352**, 712-716 (2016).
- 30. Werneburg S, *et al.* Targeted Complement Inhibition at Synapses Prevents Microglial
Synaptic Engulfment and Synapse Loss in Demyelinating Disease. *Immunity* **52**, 167-
182 e167 (2020).
- 31. Michailidou I, *et al.* Complement C1q-C3-associated synaptic changes in multiple
sclerosis hippocampus. *Ann Neurol* **77**, 1007-1026 (2015).
- 32. Paolicelli RC, *et al.* Synaptic pruning by microglia is necessary for normal brain
development. *Science* **333**, 1456-1458 (2011).
- 33. Gunner G, *et al.* Sensory lesioning induces microglial synapse elimination via
ADAM10 and fractalkine signaling. *Nat Neurosci* **22**, 1075-1088 (2019).
- 34. Nguyen PT, *et al.* Microglial Remodeling of the Extracellular Matrix Promotes
Synapse Plasticity. *Cell* **182**, 388-403 e315 (2020).
- 35. He D, *et al.* Disruption of the IL-33-ST2-AKT signaling axis impairs
neurodevelopment by inhibiting microglial metabolic adaptation and phagocytic
function. *Immunity* **55**, 159-173 e159 (2022).
- 36. Vainchtein ID, *et al.* Astrocyte-derived interleukin-33 promotes microglial synapse
engulfment and neural circuit development. *Science* **359**, 1269-1273 (2018).
- 37. Blanco-Suarez E, Caldwell AL, Allen NJ. Role of astrocyte-synapse interactions in
CNS disorders. *J Physiol* **595**, 1903-1916 (2017).
- 38. Chung WS, *et al.* Astrocytes mediate synapse elimination through MEGF10 and
MERTK pathways. *Nature* **504**, 394-400 (2013).
- 39. Auguste YSS, *et al.* Oligodendrocyte precursor cells engulf synapses during circuit

- remodeling in mice. *Nat Neurosci* **25**, 1273-1278 (2022).
- 40. Heinz DE, *et al.* Exploratory drive, fear, and anxiety are dissociable and independent
components in foraging mice. *Transl Psychiatry* **11**, 318 (2021).
- 41. Cryan JF, Holmes A. The ascent of mouse: advances in modelling human depression
and anxiety. *Nat Rev Drug Discov* **4**, 775-790 (2005).
- 42. Crawley JN. Exploratory behavior models of anxiety in mice. *Neurosci Biobehav Rev*
**9**, 37-44 (1985).
- 43. Berglund R, *et al.* Microglial autophagy-associated phagocytosis is essential for
recovery from neuroinflammation. *Sci Immunol* **5**, (2020).
- 44. Xu F, *et al.* Prolonged anesthesia induces neuroinflammation and complement-
mediated microglial synaptic elimination involved in neurocognitive dysfunction and
anxiety-like behaviors. *BMC Med* **21**, 7 (2023).
- 45. Au NPB, Ma CHE. Neuroinflammation, Microglia and Implications for Retinal
Ganglion Cell Survival and Axon Regeneration in Traumatic Optic Neuropathy. *Front*
*Immunol* **13**, 860070 (2022).
- 46. Zhou J, *et al.* The neuronal pentraxin Nptx2 regulates complement activity and restrains
microglia-mediated synapse loss in neurodegeneration. *Sci Transl Med* **15**, eadf0141
(2023).
- 47. Gehrman J, Banati RB. Microglial turnover in the injured CNS: activated microglia
undergo delayed DNA fragmentation following peripheral nerve injury. *J Neuropathol*
*Exp Neurol* **54**, 680-688 (1995).
- 48. Lawson LJ, Perry VH, Gordon S. Turnover of resident microglia in the normal adult
mouse brain. *Neuroscience* **48**, 405-415 (1992).
- 49. Askew K, *et al.* Coupled Proliferation and Apoptosis Maintain the Rapid Turnover of
Microglia in the Adult Brain. *Cell Rep* **18**, 391-405 (2017).
- 50. Asakuno K, *et al.* The exogenous control of transfected c-fos gene expression and
angiogenesis in cells implanted into the rat brain. *Brain Res* **702**, 23-31 (1995).
- 51. Yuan T, Orock A, Greenwood-Van Meerveld B. Amygdala microglia modify neuronal
plasticity via complement C1q/C3-CR3 signaling and contribute to visceral pain in a
rat model. *Am J Physiol Gastrointest Liver Physiol* **320**, G1081-G1092 (2021).
- 52. Acharjee S, *et al.* Reduced Microglial Activity and Enhanced Glutamate Transmission
in the Basolateral Amygdala in Early CNS Autoimmunity. *J Neurosci* **38**, 9019-9033
(2018).
- 53. Cao P, *et al.* Early-life inflammation promotes depressive symptoms in adolescence
via microglial engulfment of dendritic spines. *Neuron* **109**, 2573-2589 e2579 (2021).
- 54. Frank MG, Fonken LK, Watkins LR, Maier SF. Acute stress induces chronic
neuroinflammatory, microglial and behavioral priming: A role for potentiated NLRP3
inflammasome activation. *Brain Behav Immun* **89**, 32-42 (2020).
- 55. Augustine V, *et al.* Temporally and Spatially Distinct Thirst Satiation Signals. *Neuron*
**103**, 242-249 e244 (2019).
- 56. Lee S, *et al.* Chemosensory modulation of neural circuits for sodium appetite. *Nature*
**568**, 93-97 (2019).
- 57. Wohleb ES, Terwilliger R, Duman CH, Duman RS. Stress-Induced Neuronal Colony
Stimulating Factor 1 Provokes Microglia-Mediated Neuronal Remodeling and

- Depressive-like Behavior. *Biol Psychiatry* **83**, 38-49 (2018).
- 58. Zhang J, *et al.* IL4-driven microglia modulate stress resilience through BDNF-
dependent neurogenesis. *Sci Adv* **7**, (2021).
- 59. Qing H, *et al.* Origin and Function of Stress-Induced IL-6 in Murine Models. *Cell* **182**,
372-387 e314 (2020).
- 60. Taniguchi S, Elhance A, Van Duzer A, Kumar S, Leitenberger JJ, Oshimori N. Tumor-
initiating cells establish an IL-33-TGF-beta niche signaling loop to promote cancer
progression. *Science* **369**, (2020).
- 61. De Schepper S, *et al.* Perivascular cells induce microglial phagocytic states and
synaptic engulfment via SPP1 in mouse models of Alzheimer's disease. *Nat Neurosci*
**26**, 406-415 (2023).
- 62. Umpierre AD, Wu LJ. How microglia sense and regulate neuronal activity. *Glia* **69**,
1637-1653 (2021).
- 63. Kim RY, *et al.* Astrocyte CCL2 sustains immune cell infiltration in chronic
experimental autoimmune encephalomyelitis. *J Neuroimmunol* **274**, 53-61 (2014).
- 64. Cherry JD, *et al.* CCL2 is associated with microglia and macrophage recruitment in
chronic traumatic encephalopathy. *J Neuroinflammation* **17**, 370 (2020).
- 65. Harkness KA, Sussman JD, Davies-Jones GA, Greenwood J, Woodroffe MN.
Cytokine regulation of MCP-1 expression in brain and retinal microvascular
endothelial cells. *J Neuroimmunol* **142**, 1-9 (2003).
- 66. Maren S, Phan KL, Liberzon I. The contextual brain: implications for fear conditioning,
extinction and psychopathology. *Nat Rev Neurosci* **14**, 417-428 (2013).
- 67. Pace-Schott EF, Germain A, Milad MR. Effects of sleep on memory for conditioned
fear and fear extinction. *Psychol Bull* **141**, 835-857 (2015).
- 68. Fan KQ, *et al.* Stress-Induced Metabolic Disorder in Peripheral CD4(+) T Cells Leads
to Anxiety-like Behavior. *Cell* **179**, 864-879 e819 (2019).
- 69. McCall JG, *et al.* CRH Engagement of the Locus Coeruleus Noradrenergic System
Mediates Stress-Induced Anxiety. *Neuron* **87**, 605-620 (2015).
- 70. Haubensak W, *et al.* Genetic dissection of an amygdala microcircuit that gates
conditioned fear. *Nature* **468**, 270-276 (2010).
- 71. Ahrens S, *et al.* A Central Extended Amygdala Circuit That Modulates Anxiety. *J*
*Neurosci* **38**, 5567-5583 (2018).
- 72. Tye KM, *et al.* Amygdala circuitry mediating reversible and bidirectional control of
anxiety. *Nature* **471**, 358-362 (2011).
- 73. Liu L, *et al.* Cell type-differential modulation of prefrontal cortical GABAergic
interneurons on low gamma rhythm and social interaction. *Sci Adv* **6**, eaay4073 (2020).
- 74. Breton-Provencher V, Sur M. Active control of arousal by a locus coeruleus
GABAergic circuit. *Nat Neurosci* **22**, 218-228 (2019).

REVIEWER COMMENTS

Reviewer #1 (Remarks to the Author):

The authors have adequately addressed my previous concerns by adding new experimental data and discussions. To further improve the paper, it is better to add a schematic model illustrating the main conclusions.

Reviewer #2 (Remarks to the Author):

The authors have addressed some of my primary concerns, but there are remain significant issues with data interpretation.

In particular, the authors continue to use anxiety-like behavior to describe rodent behaviors.

They also maintain that morphological features of microglia provide evidence of neuroinflammation. Some immunohistology for colocalization of TNF α , IL1b, and IL6 are presented but these proteins are notoriously difficult to immunolabel. Antibodies should be tested in corresponding knockout mice or other molecular analyses should be used to validate proposed neuroinflammatory phenotype.

The authors also contend that increases in microglia are due to rapid proliferation, but Ki67 immunolabeling is suspicious and does not look confined to the nucleus. Further it appears that TUNEL immunolabeling is observed in non-microglia cells at 8 hours, which is concerning.

Other new data support some conclusions, but no orthogonal images are provided to validate colocalization of synaptic markers and lysosomes.

Also the authors provide an unconvincing arguments to why their controls had high levels of synaptic engulfment (in the original submission).

Last, the title seems problematic as it is unclear what "fading" means in the context of behavioral outcomes. Also the use of only male mice should be noted in the title.

Reviewer #3 (Remarks to the Author):

The authors have adequately addressed all my comments.

**Response to referees**

**Manuscript ID:** NCOMMS-23-10335B

**Title:** Microglia govern the extinction of acute stress-induced low level of exploratory
behaviors in male mice

We sincerely appreciate the time and efforts of the Editor and Reviewers in evaluating our study.

In light of their thoughtful critique, we have performed additional experiments to validate our

previous data and responded to each of their concerns with support from the literature. We have

also thoroughly revised the manuscript and incorporated these suggestions into the revised

manuscript where appropriate. The revised manuscript with tracked changes (highlighted in

blue) has been uploaded as a separate file. The detailed changes and our point-by-point

responses to each of the Reviewers' comments are presented below.

**Reviewer #1 (Remarks to the Author):**

The authors have adequately addressed my previous concerns by adding new experimental data
and discussions. To further improve the paper, it is better to add a schematic model illustrating
the main conclusions.

**Response:** We appreciate the Reviewer's thoughtful comments and guidance throughout the
review process. As suggested by the Reviewer, we have added a schematic model to the revised
manuscript illustrating our main conclusions (Response Figure 1 and new Supplementary
Figure 16).

**Response Figure 1. Microglial engulfment of dendritic spines promotes the extinction of**
**acute stress-induced low level of exploratory behaviors.**

Low level of exploratory behaviors following acute restraint are relieved within 12 hours after
stress induction in male mice. Suppression of NF-κB by MST4 stimulates production of
CX3CL1 by CeA^{GABA} neurons, which increases under acute restraint stress and subsequently
activates microglia in the CeA, promoting engulfment of dendritic spines. Microglial
engulfment of dendritic spines in the CeA leads to feedback inhibition that attenuates CeA^{GABA}
neuronal hyperactivity, restoring them to non-stress levels and leading to extinction of low level
of exploratory behaviors.

**Reviewer #2 (Remarks to the Author):**

The authors have addressed some of my primary concerns, but there are remain significant
issues with data interpretation.

1. In particular, the authors continue to use anxiety-like behavior to describe rodent behaviors.

**Response:** We completely agree that using the term “anxiety-like behavior” may be an
inaccurate description of the phenotype defined by EPM and OFT behavioral assays, since these
tests cannot fully capture the complexity of anxiety disorders. As suggested, we have replaced
the “anxiety-like behaviors” with “low level of exploratory behaviors” in the revised
manuscript.

2. They also maintain that morphological features of microglia provide evidence of
neuroinflammation. Some immunohistology for colocalization of TNF α , IL1b, and IL6 are
presented but these proteins are notoriously difficult to immunolabel. Antibodies should be
tested in corresponding knockout mice or other molecular analyses should be used to validate
proposed neuroinflammatory phenotype.

**Response:** Following the Reviewer’s advice, we examined neuroinflammation in the CeA
using *Cx3cr1-GFP* transgenic mice, which express a GFP label in microglial cells throughout
the brain. We then isolated CeA microglia by fluorescence-activated cell sorting (FACS), and
examined changes in *Tnf- α* , *Il-1 β* and *Il-6* transcript levels by qPCR. The results showed that
all of these inflammatory markers were expressed at significantly higher levels in ARS-2h mice
than in CeA microglia of non-stressed controls at 0.5 h post-stress treatment (**Response Figure**
**2, and new Supplementary Figure 5c-e**). We hope the Reviewer is now convinced that the
observed cellular responses are indeed associated with increased neuroinflammation.

These new results are presented in the revised manuscript.

Response Figure 2. Expression of classical inflammatory factors in microglia from the CeA of ARS-2h mice.

a, Representative images of microglia in the CeA of *Cx3cr1-GFP* mice. Scale bars, 500 μ m (left) and 50 μ m (right). **b**, Gating strategy of the cell subpopulations in the CeA GFP⁺ microglia analyzed by flow cytometry. **c**, qPCR analysis of *Tnf- α* , *Il-1 β* and *Il-6* mRNA levels in CeA microglia of ARS-2h and control mice (n = 3 samples per group). All data show mean \pm SEM. ***p* < 0.01, and ****p* < 0.001. Also see Table S1.

3. The authors also contend that increases in microglia are due to rapid proliferation, but Ki67 immunolabeling is suspicious and does not look confined to the nucleus. Further it appears that TUNEL immunolabeling is observed in non-microglia cells at 8 hours, which is concerning.

Response: We thank the Reviewer for close attention to detail and for their alternative interpretations of Ki67 and TUNEL staining. We have conducted new experiments to address the potential artifacts the Reviewer has proposed.

Regarding the “suspicious” Ki67 immunolabeling that “does not look confined to the nucleus”, we agree that these doubts warrant further consideration. Ki67 is commonly used to detect cell proliferation based on its role in the cell cycle, and therefore immunostains for Ki67 should localize to the nucleus. We examined numerous studies and product validation documents to search for possible causes of Ki67 signal outside of the nucleus and found that

although colocalization of Ki67 and nuclear signals have been reported in many studies using
 the same antibody we selected for our work (Cat# 14-5698-82, Invitrogen)^{1,2,3,4,5}, some articles
 reported incomplete colocalization with the nuclear signal^{6,7,8}. This inconsistency suggested
 that the Ki67 antibody we used might be defective or otherwise exhibit poor specificity for
 Ki67. We therefore chose another widely used Ki67 antibody (Cat# 12202, Cell Signaling
 Technology)^{9,10,11} and repeated the Ki67 immunolabeling experiments in mouse brain slices.
 The results showed strong co-labeling of Ki67 with the nucleus of Iba1⁺ microglia in CeA
 microglia of ARS-2h mice at 0.5 h and 8 h post treatment, while there were no Ki67 signals in
 control or ARS-2h mice at 12 h post treatment (Response Figure 3a, and new Supplementary
 Figure 6a). These results suggest that Ki67 immunolabeling does not look confined to the
 nucleus in our previous experiment potentially due to poor specificity of the Ki67 antibody.

Regarding the TUNEL immunolabeling observed in non-microglial cells at 8 hours post
 treatment, in our previous experiment, it is possible that: although widely used for detecting
 apoptosis in various cell types, TUNEL cannot specifically differentiate microglial apoptosis
 from apoptosis in other cells, which could be occurring in non-microglial cells in the
 CeA under restraint-induced acute stress conditions. For example, previous studies have shown
 apoptosis in microglia of multiple brain regions induced by chronic stress is accompanied by
 loss of other, non-microglia cells^{12,13}; Further, after stress occurs in mice, monocytes and other
 macrophages in the peripheral blood will enter the brain through the blood-brain barrier to
 function^{14,15}. However, it is still unclear whether these peripheral cells entering the brain also
 undergo apoptosis to restore brain homeostasis after the removal of stress stimuli.

These collective results suggest that proliferation and apoptosis of microglia likely
 contribute to changes in microglia density, further illustrating the dynamic activation of
 microglia and their subsequent restoration to resting levels in the CeA following ARS-2h
 treatment. We hope these replicate experiments, which are presented in the revised manuscript,
 allay the Reviewer's doubts about the quality of the data.

 **Response Figure 3. Immunofluorescence staining for Ki67 following ARS-2h treatment in**
 **mice.**

Representative images of immunostaining for Ki67 (red), Iba1 (green), and DAPI (blue) in the

CeA of ARS-2h and control mice at 0.5 h/8 h/12 h post-stress induction. Scale bars, 10 μ m.

4. Other new data support some conclusions, but no orthogonal images are provided to validate
colocalization of synaptic markers and lysosomes.

**Response:** We appreciate this advice. Although 3D reconstruction has been used in other
studies to effectively demonstrate the phagocytosis of microglia^{16,17,18}, we agree that orthogonal
images could provide direct and solid validation of synaptic marker colocalization with
lysosomes. As suggested, we now provide orthogonal images depicting colocalization between
immunoreactive puncta of GAD65/67, CD68, and Iba1⁺ microglia in the revised manuscript
(**Response Figure 4, and new Figure 3i**).

**Response Figure 4. Microglial engulfment of synaptic structures in ARS-2h mice at 0.5**
**h/12 h post-stress induction.**

Representative images of immunostaining for CD68 (red), Iba1 (green), and GAD65/67⁺ puncta
(purple) in the CeA of ARS-2h and control mice at 0.5 h/12 h post-stress induction. Orthogonal
images have been included. Scale bars, 5 μ m.

5. Also the authors provide an unconvincing arguments to why their controls had high levels of
synaptic engulfment (in the original submission).

**Response:** Regarding the Reviewer's concern, our argument in the "response to referees" for
Manuscript ID: NCOMMS-23-1335A is as follows:

*Regarding the Reviewer's concern that "these results are questionable as it appears*
*that even in control mice there is an unusually high number of inclusions in*
*microglia", we believe that there may be two reasons for this. First, we checked the*
*related studies of the GAD65/67 antibody (ab183999, abcam) used in this study. The*
*WB experiments of the antibody are provided by the official website of Abcam*
*([https://www.abcam.cn/products/primary-antibodies/gad65--gad67-antibody-](https://www.abcam.cn/products/primary-antibodies/gad65--gad67-antibody-epr19366-ab183999.html)*
*[epr19366-ab183999.html](https://www.abcam.cn/products/primary-antibodies/gad65--gad67-antibody-epr19366-ab183999.html)), which showed two bands with antibodies against GAD67*

*and GAD65 fragment recombinant proteins in mouse. In addition, the GAD65/67*
*antibody have been widely used in immunofluorescent staining in numerous*
*studies^{19,20}. Second, although there are many GAD65/67 punctas in microglia from*
*control mice, we found that the macrophage marker CD68 is rarely expressed in the*
*control mice (please see original Figure 3g, h), so there should not be as much*
*microglial engulfment in the control mice. Therefore, it seems that the low threshold*
*adjustment in the algorithm of the microglial engulfment analysis results in unusually*
*high number of inclusions in microglia from control mice. Briefly, the Imaris*
*MATLAB-based (MathWorks) plugin “Split into Surface Objects” was used to assess*
*the number of GAD65/67 puncta in microglia (distance $\leq 0 \mu\text{m}$). Based on this*
*analytical method, the data of the engulfed synaptic marker are from the GAD65/67*
*puncta that entirely within microglia as well as those distributed on these cell surface.*
*As suggested, we have readjusted the “Estimated XY Diameter” from $0.8 \mu\text{m}$ to 0.9*
*μm , which is used to estimate the size of GAD65/67 puncta. After adjusting the*
*parameters, we found that the number of GAD65/67 puncta in microglia from the*
*control groups and ARS-2h mice were significantly reduced, but the increase in the*
*number of GAD65/67 puncta in CeA microglia at 0.5 h post-ARS-2h compared with*
*control mice, was still remained (please see Response Document Figure 11c, d, and*
*also see new Figure 3h). These results indicate that the threshold is indeed too low.*

We appreciate the Reviewer’s efforts to ensure the rigor of our data, and we regret that
they found our previous explanation unconvincing. To more convincingly illustrate our point,
we now provide images of the control group in Figure 3h before and after adjusting the
“Estimated XY Diameter” threshold in the IMARIS software. After readjusting “Estimated XY
Diameter” from $0.8 \mu\text{m}$ to $0.9 \mu\text{m}$, which is used to estimate the size of GAD65/67 puncta, we
found that the number of GAD65/67 puncta significantly decreased in control group microglia
(Response Figure 5a, b), suggesting that the high levels of synaptic engulfment in the control
group was likely due to an excessively low threshold setting ($0.8 \mu\text{m}$). Moreover, in order to
clearly demonstrate the specific phagocytosis of microglia, we also now provide orthogonal
images validating GAD65/67 colocalization with microglia (Response Figure 5c).

The Reviewer will kindly note that we have already provided experimental data after
adjusting the threshold of size of GAD65/67 puncta to $0.9 \mu\text{m}$ in the previous round of review.
Those data clearly demonstrate that both ARS-2h and control mice have significantly fewer
GAD65/67 puncta in microglia at baseline compared to the original data generated with an 0.8
169 μm threshold, while the significant increase in GAD65/67 puncta in CeA microglia at 0.5 h
post-ARS-2h, but not control mice, could still be observed (Response Figure 5d, e, and new
Figure 3h).

**Response Figure 5. Comparison of GAD65/67⁺ puncta quantification in control microglia**
 **by IMARIS software at different “Estimated XY Diameter” thresholds.**

**a, b**, Representative images from IMARIS software of GAD65/67⁺ puncta quantification in
 CeA microglia of control mice from Figure 3h using an “Estimated XY Diameter” threshold of
 0.8 μm (**a**) and “Estimated XY Diameter” of 0.9 μm (**b**). **c**, Example orthogonal image without
 spots after adjusting parameters. The position of this field of view is indicated by the white
 arrow in (**a**). Scale bars, 2 μm . **d**, Representative images and 3D surface rendering of Iba1⁺
 microglia (green) containing GAD65/67⁺ puncta (red) in the CeA of ARS-2h and control mice
 at 0.5 h/12 h post-stress induction. Scale bars, 50 μm (overview) and 10 μm (inset and
 rendering). **e**, Summary of GAD65/67⁺ puncta quantification in microglia of mice from (**d**) (n
 = 53 cells from six mice per group). All data show mean \pm SEM. *** $p < 0.001$, n.s., not
 significant. Also see Table S1.

6. Last, the title seems problematic as it is unclear what “fading” means in the context of
 behavioral outcomes. Also the use of only male mice should be noted in the title.

**Response:** We agree with this constructive assessment. Although we thought “extinction”
 might be appropriate in the original manuscript, we were concerned that is most commonly
 used in psychology-related studies, such as research exploring the neural mechanisms of
 memory and fear^{21,22}. Upon further consideration, we believe extinction may connote a meaning
 consistent with the dynamic behavioral changes observed in our study. As recommended, we
 have replaced “fading” with “extinction” in the most recent version of the manuscript, and we

have added the term “male mice” to comply with journal policy. After careful consideration of
the Reviewer’s comments, we have changed the title to **Microglia govern the extinction of**
**acute stress-induced low level of exploratory behaviors in male mice.**

We again thank the Reviewer for constructive critique towards ensuring the rigor of our
experiments. We hope the Reviewer now finds our evidence sufficiently convincing that we
have indeed found a bona fide phenomenon of microglial response to acute stress.

**Reviewer #3 (Remarks to the Author):**

The authors have adequately addressed all my comments.

**Response:** We appreciate the Reviewer's thoughtful comments throughout the review process.

**References**

- 1. Soucie EL, *et al.* Lineage-specific enhancers activate self-renewal genes in
macrophages and embryonic stem cells. *Science* **351**, aad5510 (2016).
- 2. Kamizaki K, *et al.* The Ror1 receptor tyrosine kinase plays a critical role in regulating
satellite cell proliferation during regeneration of injured muscle. *J Biol Chem* **292**,
15939-15951 (2017).
- 3. Casanova-Acebes M, *et al.* Tissue-resident macrophages provide a pro-tumorigenic
niche to early NSCLC cells. *Nature* **595**, 578-584 (2021).
- 4. Denoth-Lippuner A, *et al.* Visualization of individual cell division history in complex
tissues using iCOUNT. *Cell Stem Cell* **28**, 2020-2034.e2012 (2021).
- 5. Yamaguchi N, *et al.* Voluntary running exercise modifies astrocytic population and
features in the peri-infarct cortex. *IBRO Neurosci Rep* **14**, 253-263 (2023).
- 6. O'Connell JS, *et al.* Treatment of necrotizing enterocolitis by conditioned medium
derived from human amniotic fluid stem cells. *PLoS One* **16**, e0260522 (2021).
- 7. Koren E, *et al.* ARTS mediates apoptosis and regeneration of the intestinal stem cell
niche. *Nat Commun* **9**, 4582 (2018).
- 8. Zondervan RL, Jenkins DC, Reicha JD, Hankenson KD. Thrombospondin-2
spatiotemporal expression in skeletal fractures. *J Orthop Res* **39**, 30-41 (2021).
- 9. Wang Z, *et al.* Binding of PLD2-Generated Phosphatidic Acid to KIF5B Promotes
MT1-MMP Surface Trafficking and Lung Metastasis of Mouse Breast Cancer Cells.
*Dev Cell* **43**, 186-197.e187 (2017).
- 10. Kim BH, Jung HW, Seo SH, Shin H, Kwon J, Suh JM. Synergistic actions of FGF2 and
bone marrow transplantation mitigate radiation-induced intestinal injury. *Cell Death*
*Dis* **9**, 383 (2018).
- 11. Metzger R, Maruskova M, Krebs S, Janssen KP, Krug AB. Increased Incidence of
Colon Tumors in AOM-Treated Apc (1638N/+) Mice Reveals Higher Frequency of
Tumor Associated Neutrophils in Colon Than Small Intestine. *Front Oncol* **9**, 1001
(2019).
- 12. Guo J, *et al.* Microglia Loss and Astrocyte Activation Cause Dynamic Changes in
Hippocampal [(18)F]DPA-714 Uptake in Mouse Models of Depression. *Front Cell*
*Neurosci* **16**, 802192 (2022).
- 13. Su D, *et al.* Chronic exposure to aflatoxin B1 increases hippocampal microglial
pyroptosis and vulnerability to stress in mice. *Ecotoxicol Environ Saf* **258**, 114991
(2023).
- 14. Wohleb ES, Powell ND, Godbout JP, Sheridan JF. Stress-induced recruitment of bone
marrow-derived monocytes to the brain promotes anxiety-like behavior. *J Neurosci* **33**,
13820-13833 (2013).
- 15. Yu X, *et al.* Extracellular vesicle-mediated delivery of circDYM alleviates CUS-
induced depressive-like behaviours. *J Extracell Vesicles* **11**, e12185 (2022).

- 16. Anderson SR, *et al.* Complement Targets Newborn Retinal Ganglion Cells for
Phagocytic Elimination by Microglia. *J Neurosci* **39**, 2025-2040 (2019).
- 17. Lee JY, *et al.* Neuronal SphK1 acetylates COX2 and contributes to pathogenesis in a
model of Alzheimer's Disease. *Nat Commun* **9**, 1479 (2018).
- 18. Fu AK, *et al.* IL-33 ameliorates Alzheimer's disease-like pathology and cognitive
decline. *Proc Natl Acad Sci U S A* **113**, E2705-2713 (2016).
- 19. Augustine V, *et al.* Temporally and Spatially Distinct Thirst Satiation Signals. *Neuron*
**103**, 242-249 e244 (2019).
- 20. Lee S, *et al.* Chemosensory modulation of neural circuits for sodium appetite. *Nature*
**568**, 93-97 (2019).
- 21. Maren S, Phan KL, Liberzon I. The contextual brain: implications for fear conditioning,
extinction and psychopathology. *Nat Rev Neurosci* **14**, 417-428 (2013).
- 22. Pace-Schott EF, Germain A, Milad MR. Effects of sleep on memory for conditioned
fear and fear extinction. *Psychol Bull* **141**, 835-857 (2015).

REVIEWERS' COMMENTS

Reviewer #2 (Remarks to the Author):

The authors have provided relevant details and data to support their conclusions. I appreciate their efforts to address my prior reviews. I remain concerned about low sample sizes ($n < 5$) for several studies, as well as the rigor and interpretation of approaches to examine microglia morphology, turnover (proliferation and apoptosis), and synaptic engulfment. In particular, the authors should note the potential limitations and technical issues uncovered in response to my prior comments (i.e., non-specific antibodies). This is important as it can lead to spurious conclusions about how microglia shape neurobiology. Specifically that autofluorescence in microglia can be mistaken for "antibody-labeled material (see Stillman et al, biorxiv, 2023).

**Response to referees**

**Manuscript ID:** NCOMMS-23-10335C

**Title:** Microglia govern the extinction of acute stress-induced anxiety-like behaviors in male
mice

We sincerely appreciate the time and efforts of the Editor and Reviewers in evaluating our study.

As suggested, we have thoroughly revised the manuscript and incorporated these suggestions

into the revised manuscript where appropriate. The revised manuscript with tracked changes

(highlighted in blue) has been uploaded as a separate file. The detailed changes and our point-

by-point responses to each of the Reviewers' comments are presented below.

**Reviewer #2:**

The authors have provided relevant details and data to support their conclusions. I appreciate
their efforts to address my prior reviews.

1. I remain concerned about low sample sizes ($n < 5$) for several studies, as well as the rigor and
interpretation of approaches to examine microglia morphology, turnover (proliferation and
apoptosis), and synaptic engulfment. In particular, the authors should note the potential
limitations and technical issues uncovered in response to my prior comments (i.e., non-specific
antibodies). This is important as it can lead to spurious conclusions about how microglia shape
neurobiology. Specifically that autofluorescence in microglia can be mistaken for "antibody-
labeled material (see Stillman et al, biorxiv, 2023).

**Response:** We greatly appreciate these constructive comments.

In our study, three samples were only used in the experiments for qPCR detection of
inflammatory genes for each group after fluorescence-activated cell sorting of CeA microglia
in this study (Supplementary Fig. 5). It should be pointed out that the minimum number of cells
used for qPCR should not be less than 10^6 , in order to meet this experimental requirement each
sample here refers to the total number of CeA microglia obtained from 20 mice, which is due
to the small CeA brain region of mice. Moreover, since the significant differences have already
been developed on the basis of the three samples for each inflammatory gene between control
and ARS-2h mice (*Tnf- α* , $p = 0.0090$; *Il-1 β* , $p < 0.0001$; *Il-6*, $p = 0.0094$), we did not further
increase more sample size.

Regarding the Reviewer's concern about the specificity of antibodies, this is indeed a very
general issue in the field, and we have provided a detailed explanation to this issue in the last
"Response to referees". In addition, as mentioned by the Reviewer, a recent study has reported
that the autofluorescence of lipofuscin can be likely detected within microglial lysosomes in
the adult mouse brain by light microscopy¹. To address this general issue in the field, we used
the same aged mice as corresponding controls to detect microglial engulfment throughout our
study, which was able to minimize the possibility that autofluorescence could be mistaken for
an "antibody-labeled material" affecting statistical differences.

Considering the potential impact of the specificity of antibodies and autofluorescence on
the conclusions raised by the Reviewers, we have highlighted these limitations in the discussion
of the revised manuscript.

We thank the Reviewer for the helpful guidance about how to improve our study.

**References**

- 1. Stillman JM, Mendes Lopes F, Lin JP, Hu K, Reich DS, Schafer DP. Lipofuscin-like
autofluorescence within microglia and its impact on studying microglial engulfment.